# Satellite derived leaf area index and roughness length information for surface-atmosphere exchange modelling: a case study for reactive nitrogen deposition in north-western Europe using LOTOS-EUROS v2.0.

Shelley C. van der Graaf[1], Richard Kranenburg[2], Arjo J. Segers[2], Martijn Schaap[2,3], Jan Willem Erisman[1,4]

[1]Cluster Earth and Climate, Department of Earth Sciences, Faculty of Science, Vrije Universiteit Amsterdam, Amsterdam, 1081 HV, The Netherlands

[2]TNO, Climate Air and Sustainability, Utrecht, 3584 CB, The Netherlands

[3]Institute for Meteorology, Free University Berlin, Berlin, 12165, Germany

[4]Louis Bolk Institute, Driebergen, 3972, The Netherlands

*Correspondence to:* Shelley C. van der Graaf (s.c.vander.graaf@vu.nl)

**Abstract.** The nitrogen cycle has been continuously disrupted by human activity over the past century, resulting in almost a tripling of the total reactive nitrogen fixation in Europe. Consequently, excessive amounts of reactive

nitrogen ($N_r$) have manifested in the environment, leading to a cascade of adverse effects, such as acidification and eutrophication of terrestrial and aquatic ecosystems, and particulate matter formation. Chemistry transport models (CTM) are frequently used as tools to simulate the complex chain of processes that determine atmospheric $N_r$ flows. In these models, the parameterization of the atmosphere-biosphere exchange of $N_r$ is largely based on few surface exchange measurement and is therefore known to be highly uncertain. In addition to this, the input parameters that

are used here are often fixed values, only linked to specific land use classes. In an attempt to improve this, a combination of multiple satellite products is used to derive updated, time-variant leaf area index (LAI) and roughness length ($z_0$) input maps. As LAI, we use the MODIS MCD15A2H product. The monthly $z_0$ input maps presented in this paper are a function of satellite-derived NDVI values (MYD13A3 product) for short vegetation types (such as grass and arable land) and a combination of satellite-derived forest canopy height and LAI for forests.

The use of these growth-dependent satellite products allows us to represent the growing season more realistically. For urban areas, the $z_0$ values are updated, too, and linked to a population density map. The approach to derive these dynamic $z_0$ estimates can be linked to any land use map and is as such transferable to other models. We evaluated the sensitivity of the modelled $N_r$ deposition fields in LOTOS-EUROS v2.0 to the abovementioned changes in LAI and $z_0$ inputs, focusing on  Germany, the Netherlands and Belgium. We computed $z_0$ values from FLUXNET sites

and compared these to the default and updated $z_0$ values in LOTOS-EUROS. The RMSD for both short vegetation and forest sites improved. Comparing all sites, the RMSD decreased from 0.76 (default $z_0$) to 0.60 (updated $z_0$). The implementation of these updated LAI and $z_0$ input maps led to local changes in the total $N_r$ deposition of up to ~30%

and a general shift from wet to dry deposition. The most distinct changes are observed in land use specific deposition fluxes. These fluxes may show relatively large deviations, locally affecting estimated critical load

exceedances for specific natural ecosystems.

## 1. Introduction

The nitrogen (N) cycle has been continuously disrupted by human activity over the past century (Fowler et al., 2015;Galloway et al., 2004;Galloway et al., 2008), resulting in a doubling of the total reactive nitrogen ($N_r$) fixation globally and even a tripling in Europe. As a result, excessive amounts of $N_r$, defined as all N compounds except $N_2$,

have manifested in the environment contributing to acidification and eutrophication of sensitive terrestrial and aquatic ecosystems (Bobbink et al., 2010a;Paerl et al., 2014). $NO_x$ and $NH_3$ affect air quality through their significant role in the formation of particulate matter, impacting human health and life expectancy (Lelieveld et al., 2015;Bauer et al., 2016;Erisman and Schaap, 2004). $N_r$ also influences climate change through its impact on greenhouse gas emissions and radiative forcing (Erisman et al., 2011;Butterbach-Bahl et al., 2011). As $N_r$ forms are

linked through chemical and biological conversion in one another within the environmental compartments, one atom of N may even take part in a cascade of $N_r$ forms and effects (Galloway et al., 2003). To minimize these adverse effects, effective nitrogen management and policy development, therefore, require consideration of all $N_r$ forms simultaneously.

With the scarceness and inadequate distribution of available ground measurements, especially for reduced $N_r$, the

most important method to assess and quantify total $N_r$ budgets on a larger spatial scale to date remains the use of models.  Models, chemistry transport models, in particular, are used for understanding the atmospheric transport and the atmosphere-biosphere exchange of nitrogen compounds. Most chemistry transport models compare reasonably with observations for oxidized forms of $N_r$, but need improvement when it comes to the reduced forms of $N_r$ (Colette et al., 2017). Modelled $NH_3$ fields are in general uncertain due to the highly reactive nature and the

uncertain lifetime of $NH_3$ in the atmosphere. More importantly, $NH_3$ emissions that are used as model input are very complex to estimate and remain highly uncertain (Reis et al., 2009;Behera et al., 2013), for example, due to the diversity in $NH_3$ volatilization rates originating from different agricultural practices. Recently, a lot of effort has been made to improve the spatiotemporal distributions of bottom-up $NH_3$ emissions (e.g. (Hendriks et al., 2016;Skjøth et al., 2011)). Emissions can also be estimated top-down through the usage of data assimilation and

inversion techniques. Optimally combining observations and chemistry transport models have already enabled us to create large-scale emission estimates for various pollutants (e.g. (Curier et al., 2014;Abida et al., 2017)), for instance for $NO_2$, and will likely also be used for large-scale $NH_3$ emission estimates in the future.

Most data assimilation and inversion methods rely on the assumption that sink terms in the model hold a negligible uncertainty. To obtain reasonable top-down emission estimates, we must thus also aim to reduce the uncertainty

involved on this side of models. The sink strengths of trace gases and particles in chemistry transport models are often pragmatic and computed with relatively simple empirical functions (e.g. following (Wesely, 1989;Emberson et al., 2000;Erisman et al., 1994)), mostly linked to land use classification maps. The parameterization of the

atmosphere-biosphere exchange of $N_r$ components that is used in models is largely based on surface exchange measurements and is therefore very uncertain, especially for $NH_3$ (Schrader et al., 2016). The deposition strengths in models may vary tremendously depending on the used deposition parameterisation and velocities (Wu et al., 2018;Schrader and Brummer, 2014;Flechard et al., 2014). Moreover, inter-model discrepancies in deposition fluxes may also arise from differences in the used input variables. Here, we focus on the leaf area index (LAI) and the roughness length ($z_0$) input values. The deposition velocity is often parameterized using both the LAI and the $z_0$. Currently, most models use fixed, land use specific values for both parameters. In practice, however, spatial as well as seasonal variation is observed. In this paper, we aim to improve the deposition flux modelling by using more realistic, spatial- and time-variant LAI and $z_0$ values that are derived from optical remote sensors.

The LAI is defined as the one-sided green leaf area per unit surface area (Watson, 1947). The LAI serves as a measure for the amount of plant canopy, and herewith directly related to energy and mass exchange processes. As a result, the LAI is nowadays used as one of the main parameters in many ecological models. In deposition modelling, stomatal uptake is often parameterised using the LAI. The LAI can be determined in the field using either direct methods, such as leaf traps, or indirect methods, such as hemispherical photography (Jonckheere et al., 2004). For larger areas, the LAI can be simulated using land surface or biosphere models. Another group of indirect methods to estimate the LAI for large regions is the use of optical remote sensing. The LAI can, for instance, be estimated using empirical relationships between LAI and vegetation indices (e.g. (Soudani et al., 2006;Davi et al., 2006;Turner et al., 1999)) or by inversion of canopy reflectance models (e.g. (Houborg and Boegh, 2008;Myneni et al., 2015)). A well-known example of the latter is the LAI product from the MODIS instrument, which we will use in this study.

The $z_0$ is used to describe the surface roughness. The surface roughness serves as a momentum sink for atmospheric flow and is, therefore, an important term in atmospheric modelling. The interaction between the boundary layer and the roughness of the Earth's surface results in shear stress that affects the wind speed profile. Under neutral conditions the resulting wind profile can be approximated using a logarithmic profile:

$$U(z) = \frac{u_*}{k} ln \left(\frac{z}{z_0}\right)$$
(eq. 1)

$U(z)$ represents the mean wind speed, $u_*$ the friction velocity and $k$ the Von Kármán constant. Here, $z_0$ is a constant that represents the height at which the wind speed theoretically becomes zero. The $z_0$ can be estimated from in-situ wind speed measurements using bulk transfer equations. More recently, several studies have shown that $z_0$ for specific, uniform land cover types can also be estimated from optical remote sensing measurements, for instance using vegetation indices (e.g. (Xing et al., 2017;Yu et al., 2016;Bolle and Streckenbach, 1993;Hatfield, 1988;Moran, 1990)). The $z_0$ can also be estimated using (satellite-derived) vegetation height (e.g. (Raupach, 1994;Plate, 1971;Brutsaert, 2013;Schaudt and Dickinson, 2000)).

The use of optical remote sensing data holds promising potential for improvements of the representativeness of the surface characterization in chemistry transport models. Here, we present an approach to derive monthly $z_0$ input maps using satellite-derived NDVI values (MYD13A3 product) for short vegetation types and a combination of

satellite-derived forest canopy height and LAI for forests. We validate these $z_0$ values by comparing them to $z_0$ values computed from FLUXNET observations. We also update the $z_0$ values for urban areas, using a population density map. We use the updated $z_0$ values, as well as the MODIS-LAI, as input in LOTOS-EUROS to illustrate the

effect on transport and deposition modelling of $N_r$ components. We evaluate the sensitivity of the $N_r$ deposition fields to these input parameters, focusing on Germany, the Netherlands and Belgium. Moreover, we quantify and present the implications for land use specific fluxes on a model subpixel level. Also, we compare our model outputs with wet deposition measurements of $NH_4^+$ and $NO_3^-$ and surface concentration measurements of $NH_3$ and $NO_2$.

## 2.    Model and datasets

### 2.1. LOTOS-EUROS

### 2.1.1.    Model description

The LOTOS-EUROS model is a Eulerian chemistry transport model that simulates air pollution in the lower troposphere (Manders et al., 2017). In this study the horizontal resolution is set to 0.125° by 0.0625°, corresponding to pixels of approximately 7 by 7 kilometres in size. The model uses a five-layer vertical grid that extends up 5 km

above sea level, starting with a surface layer with a fixed height of 25 meters. The next layer is a mixing layer, followed by two time-varying dynamic reservoir layers of equal thickness, and a top layer up to 5 km. LOTOS-EUROS follows the mixed layer approach and performs hourly results using ECMWF meteorology (European Centre for Medium-Range Weather Forecasts, 2016). The gas-phase chemistry uses the TNO CBM-IV scheme (Schaap et al., 2009) and the anthropogenic emissions from the TNO-MACC-III emission database (Kuenen et al.,

2014). The wet deposition parameterisation is based on the CAMx approach, and includes both in-cloud and below-cloud scavenging (Banzhaf et al., 2012). LOTOS-EUROS makes use of the CORINE/Smiatek land use map to determine input values for surface variables.

### 2.1.2.    Dry deposition

The dry deposition flux of gases is computed following the resistance approach, in which the exchange velocity $V_d$

is equal to the reciprocal sum of the aerodynamic resistance $R_a$, the quasi-laminar boundary layer resistance $R_b$ and the canopy resistance $R_c$:

$$V_d = \frac{1}{R_a + R_b + R_c}$$        (eq. 2)

$R_a$ and $R_b$ are both influenced by the wind profile, which is computed with (eq. 1). The wind profile, in turn, depends on roughness length $z_0$ associated with different land use classes. The aerodynamic resistance $R_a$ is

computed as follows:

$$R_a = \int_{z_0}^{h} \frac{\Phi\left(\frac{z}{L}\right)}{\kappa u_{*z}} dz$$        (eq. 3)

Here, $h$ is the canopy height, which is pre-defined per land use class. The empirical function $\Phi\left(\frac{z}{L}\right)$ is taken from Businger et al. (1971) and depends on the state of the atmosphere. Depending on the value of Monin-Obukhov length $L$, the following equations are used:

For a stable atmosphere $(L > 0)$:         $\Phi_s\left(\frac{z}{L}\right) = 1 + 4.7\left(\frac{z}{L}\right)$         (eq. 4)

For an unstable atmosphere $(L < 0)$:         $\Phi_u\left(\frac{z}{L}\right) = \left(1 - 15\left(\frac{z}{L}\right)\right)^{-0.25}$         (eq. 5)

For a neutral atmosphere is it equal to unity. The Monin-Obukhov length $L$ is dependent on $z_0$, and is determined as follows:

$\frac{1}{L} = S\left(a_1 + a_2 S^2\right) z_0^{(b_1 + b_2 |S| + b_3 S^2)}$         (eq. 6)

Here, $a_1, a_2, b_1, b_2$ and $b_3$ are constants (0.004349, 0.003724, -0.5034, 0.2310 and -0.0325 respectively) and $S = -0.5\left(3.0 - 0.5\, u_s |CE|\right)$ with near-surface wind speed $u_s$ and exposure factor $CE$ , depending on cloud cover and solar zenith angle. The wind speed at a reference height (10 meters) is used to obtain the friction velocity $u_*$:

$u_* = \kappa u_{10m} / \ln\left(\frac{z_{10m}}{z_0}\right)$         (eq. 7)

The quasi-laminar boundary layer resistance $R_b$ is a function of the cross-wind leaf dimension $L_d$ and the wind
speed at canopy top, $u(h)$, following the parameterisation presented in McNaughton and Van Den Hurk, 1995:

$R_b = 1.3 * 150 * \sqrt{\frac{L_d}{u(h)}}$         (eq. 8)

$L_d$ is set to 0.02 m for arable land and permanent crops, and to 0.04 m for deciduous and coniferous forests. For other land use classes $L_d$, and subsequently, the $R_b$, is equal to zero. The canopy resistance $R_c$ is computed using the DEPAC3.11 (Deposition of Acidifying Compounds) module (van Zanten et al., 2010). $R_c$ is a parallel system of
the resistances of three different pathways, the external leaf surface or cuticular resistance $R_w$, the effective soil resistance $R_{soil,eff}$ and the stomatal resistance $R_s$, and is defined as:

$R_c = \left(\frac{1}{R_w} + \frac{1}{R_{soil,eff}} + \frac{1}{R_s}\right)^{-1}$         (eq. 9)

The external leaf surface resistance $R_w$ is a function of the surface area index (SAI) and the relative humidity. The SAI is a function of the LAI. The effective soil resistance $R_{soil,eff}$ is the sum of the in-canopy resistance $R_{inc}$ and
the soil resistance $R_{soil}$. Soil resistance $R_{soil}$ has a fixed value, depending on the land use class and conditions (frozen, wet or dry). In case of $u_* > 0$ the in-canopy resistance for arable land, permanent crops and forest is computed as follows:

$$R_{inc} = \frac{14 \, h \, SAI}{u_*} \tag{eq. 10}$$

For $u_* < 0$ and other land use classes a fixed value is used. The stomatal conductance for optimal conditions is the product of the LAI and the maximum leaf resistance from (Emberson et al., 2000). This maximum value is reduced by correction factors for photoactive radiation, temperature and vapor pressure deficit to obtain the stomatal conductance $G_s$. The $R_s$ is then equal to $1/G_s$. The resistance parameterizations differ with land us type. A total of nine different land use classes are used in DEPAC. LOTOS-EUROS uses a fixed $z_0$ value for each of these land use classes. The default LAI values are also linked to the DEPAC classes, and follow a fixed temporal behaviour that describes the growing season of each land use class (Emberson et al., 2000). The bi-directional exchange of $NH_3$ is included in the implementation of the DEPAC3.11 module (Wichink Kruit et al., 2012), allowing emissions of $NH_3$ under certain atmospheric conditions. More information about the most recent version of the model can be found in Manders et al. (2017).

**2.2. Datasets**

The following section gives a short description of all the datasets that are used in this study. Firstly, a description of the LAI dataset is given. Subsequently, the datasets that are used to derive the updated $z_0$ maps are described. Finally, the in-situ observations used for validation are discussed in the last paragraphs.

**2.2.1.  MCD15A2H Leaf Area Index**

The satellite-derived Leaf Area Index (LAI) is a combined product of the MODIS instruments on board the Terra and Aqua satellites (Myneni et al., 2015). The LAI algorithm compares bidirectional spectral reflectances observed by MODIS to values evaluated with a vegetation canopy radiative transfer model that are stored in a look-up table. The algorithm then archives the mean and the standard deviation of the derived LAI distribution functions. We used the 6[th] version of the product, MCD15A2H, which has a temporal resolution of 8 days and a spatial resolution of 500 meters.

**2.2.2.  MYD13A3 NDVI**

The Normalized Difference Vegetation Index (NDVI) is a vegetation index computed with reflectances observed by the MODIS instrument on board of the Aqua satellite (Didan, 2015). The NDVI is the normalized transform of the near infrared to the red reflectance and is expressed by:

$$\text{NDVI} = \frac{\rho_{\text{NIR}} - \rho_{\text{red}}}{\rho_{\text{NIR}} + \rho_{\text{red}}} \tag{eq. 3}$$

We used the MYD13A3 product, which is the monthly NDVI product with a spatial resolution of 1 km.

**2.2.3.  Forest canopy height**

The forest canopy height is derived from LIDAR (Light Detection And Ranging) data acquired by the GLAS (Geoscience Laser Altimeter System) instrument aboard the ICESat (Ice, Cloud, and land Elevation Satellite) satellite (Zwally et al., 2002). This instrument was an altimeter that transmitted a light pulse of 1024 nm and

recorded the reflected waveform. We used the global forest canopy height product developed by Simard et al. (2011), which has a spatial resolution of 1 km.

### 2.2.4.    Population density map

The population density grid used in this study available for all European countries and provided by the European Environmental Agency (Gallego, 2010). The population density is disaggregated with the CORINE Land Cover

inventory of 2000, using data on population per commune. The resulting population density grid is downscaled to a spatial resolution of 100 meters.

### 2.2.5.    CORINE Land Cover

The CORINE Land Cover (CLC) is a land use inventory that consists of 44 classes (European Environmental Agency, 2014). The CLC is produced by computer-assisted visual interpretation of a collection of high resolution

satellite images. The CLC has a minimum mapping unit of 25 ha and a thematic accuracy of >85%. We used the latest version of the product, CLC 2012, in this study.

### 2.2.6.    In-situ reactive nitrogen observations

The modelled $NH_4^+$ and $NO_3^-$ wet deposition fluxes are compared to observations of wet-only samplers. We used observation from the Dutch LMRe network (Van Zanten et al., 2017) and the German Lander network  (Schaap et

al., 2017).The location of the stations can be found in Figure 16. The modelled $NH_3$ surface concentrations are compared to observation from the Dutch MAN network (Lolkema et al., 2015) and the European EMEP network (EMEP, 2016). The modelled $NO_2$ surface concentrations are compared to observation from AirBase (EEA, 2019). We only used background stations. The location of these stations can be found in Figure 14.

### 2.2.7.    Eddy covariance data

FLUXNET is a global network of micrometeorological towers  that measure biosphere-atmosphere exchange fluxes using the eddy covariance (EC) method. We used half-hourly observations from the FLUXNET2015 dataset (Pastorello et al., 2017) to validate the $z_0$ values. We use observations of the mean wind speed, the friction velocity, the sensible heat flux, precipitation and  air temperature $T_a$ to determine $z_0$ values . The sites used in this study are shown in Table 1.

## 3.    Methodology
### 3.1. Updated $z_0$ maps

The updated $z_0$ maps are a composition of $z_0$ values derived using different methods. We distinguish three different main approaches: 1) $z_0$ values that depend on forest canopy height, 2) $z_0$ values that depend on the NDVI and 3) new

$z_0$ values for urban areas that depend on the population density map. In addition to these three approaches, the $z_0$ values of some urban classes were set to new default values. An overview of the datasets that are used for each DEPAC land use class is given in Table 2.

The MODIS NDVI, the MODIS LAI and the GLAS forest canopy height had to be pre-processed and homogenized in order to obtain consistent input maps that can be read into the LOTOS-EUROS model. To achieve this, we created input maps for each DEPAC class on the coordinate grid of the CORINE/Smiatek land use map in LOTOS-EUROS.

First of all, the original datasets were re-projected to geographic coordinates. The following approach is used to deal with the different horizontal resolutions of the datasets: We used the CLC2012 map, having the highest horizontal resolution, as a basis for the computation of the updated $z_0$ values. For each of the other datasets, we first computed the percentages of each CORINE land cover class within every pixel. We define homogeneous pixels, consisting of nearly one CORINE land cover class, for which we will use a threshold value of 85% of the pixels area. Then we isolate and use only these (nearly) homogeneous pixels to compute $z_0$ values for each CORINE land cover class. The methods that were applied are described in the subsequent section.

### 3.1.1.  Forest canopy height derived $z_0$ values

The forest canopy height dataset derived from GLAS LiDAR observations is used to compute the $z_0$ values for each CLC2012 forest land cover class (broad-leaved forest, coniferous forest and mixed forest) that corresponds to one of the DEPAC forest land use classes (4: coniferous forest and 5: deciduous forest). Several publications relate vegetation height to $z_0$ (e.g. (Raupach, 1994;Plate, 1971;Brutsaert, 2013)). Here we used the often used equation from (Brutsaert, 2013):

$$z_0 = 0.136 * h \qquad \text{(eq. 4)}$$

The vegetation height is the most important parameter influencing turbulence near the surface, and for this reason, the used parameterisation gives a reasonable estimate of $z_0$, even though it ignores many other aspects that influence $z_0$ (e.g. density, vertical distribution of foliage). Multiple studies have illustrated that there is a seasonal variation in $z_0/h$ for different types of forests (e.g. (Yang and Friedl, 2003;Nakai, 2008)). The $z_0$ of deciduous trees is therefore additionally linked to the leaf area index to account for changes in tree foliage. The following formula is used to compute the monthly $z_0$ value for each deciduous forest pixel:

$$z_0(LAI) = z_{0,min} + \frac{LAI - LAI,min}{LAI,max - LAI,min}(z_{0,max} - z_{0,min}) \qquad \text{(eq. 5)}$$

Here the maximum roughness length $z_{0,max}$ is set to the LiDAR-derived $z_0$ from (eq. 4). The minimum roughness length $z_{0,min}$ represents the $z_0$ of leafless deciduous trees. Following the dependence of $z_0/h$ on LAI presented in Nakai (2008) and Yang and Friedl (2003), we set the $z_{0,min}$ to 80% of $z_{0,max}$.

### 3.1.2. NDVI derived $z_0$ values

Table 3 gives an overview of several studies that relate the $z_0$ value to the NDVI. The functions are derived for different vegetation types under specific conditions. Equations 6 to 12 are derived for different types of agricultural land. These equations are all within a reasonable range from one another for NDVI values below ~0.8. Therefore, we chose to use the average function of eq. 6 to eq. 11 to compute $z_0$ values for all CLC subcategories of the following DEPAC classes: "arable", "other" and "permanent crops". Figure 1 is a visualization of eq. 6 to 11 and the used mean function. Figure S1 shows histogram of all NDVI values in our study area in 2014. We computed that 7.4% of all NDVI values have a NDVI > 0.8, 1.3% have a NDVI > 0.85 and only 0.04% have a NDVI > 0.9. Virtually all NDVI values thus fall within the range where the average function does not differ much from the individual functions. The $z_0$ value of grasslands is in general lower than other vegetation types. The last equation, eq. 12, is specifically derived for grassland and is therefore used for all CLC subcategories that fall under the DEPAC class "grass".

### 3.1.3. $z_0$ values for urban areas

The default $z_0$ of urban areas in LOTOS-EUROS was set to 2 meters. We have updated the $z_0$ values for urban areas to avoid possible overestimation of $z_0$ in sparsely populated urban areas. The updated $z_0$ values for CLC2012 class 1 and 2, '1: continuous urban fabric' and '2: discontinuous urban fabric' are time-invariant and coupled to the EEA population density map. The $z_0$ values are set to 2 metres in areas with a population density higher than 5000 inhabitants/km$^2$ and to 1 metre in areas with a population density lower than 5000 inhabitants/km$^2$. The $z_0$ values of the other urban subcategories, CLC2012 class 3 to 9, are updated to the proposed values for CLC classes in (Silva et al., 2007). Figure S2 shows the resulting updated $z_0$ values for urban areas.

### 3.2. LAI and $z_0$ in LOTOS-EUROS

After the computation of the $z_0$ values, the maps for each CORINE land cover class were merged and converted into DEPAC classes using a pre-defined conversion table. As multiple CORINE land cover may translate to one single DEPAC class, the weighted average based on the respective percentage of each CORINE land cover class was computed for each pixel. We then used linear interpolation to obtain continuous $z_0$ maps for each DEPAC class. Finally, the maps were re-gridded unto the CORINE/Smiatek grid and saved into one file per month.

The default parameterization of the LAI in LOTOS-EUROS is replaced by the MCD15A2H LAI product from MODIS. First, we applied a coordinate transformation to obtain the data in geographical coordinates. The data was then re-gridded unto the grid of the CORINE/Smiatek land use map using linear interpolation. The quality tags were evaluated to identify pixels with default fill values from the MCD15A2H product. These fill-values were replaced by the default LAI values in LOTOS-EUROS, to avoid modelling discrepancies resulting from sudden jumps in LAI values. Finally, the values were sorted per DEPAC land use class and individual fields were created for each class as new input for LOTOS-EUROS.

### 3.3. $z_0$ values from EC measurements

We used the regression method (e.g. Graf et al., 2014. Chen et al., 2015) to compute $z_0$ from several eddy covariance sites. A description of the methodology and the data processing is given in this section. The wind profile in the surface layer can be approximated by:

$$\ln\left(\frac{z-d}{z_0}\right) = \frac{k\,u(z)}{u^*} + \Psi_m\left(\frac{z-d}{L}\right) \qquad\qquad \text{(eq. A)}$$

here, $z$ is the measurement height, $d$ is the displacement height, $z_0$ is the aerodynamic roughness length, $k$ is the Von-Karman constant (=0.4), $u(z)$ is the average wind speed, $u^*$ is the friction velocity and $\Psi_m$ is the integrated universal momentum function, also known as the stability correction term. $\Psi_m$ is a function of $L$, the Monin-Obukhov length, which is defined as (e.g. Erisman and Duyzer, 1991):

$$L = -\frac{u_*^3\,T_a\rho c_p}{kgH} \qquad\qquad \text{(eq. A)}$$

where $T_a$ is the air temperature, $\rho$ the air density (= 1.2 kg m$^{-3}$), $c_p$ the heat capacity at constant pressure (=1005 J kg$^{-1}$ K$^{-1}$), $g$ the acceleration due to gravity, and $H$ the sensible heat flux. Stability correction term $\Psi_m$ is in principle a non-linear function, however, for a certain stability range it can be approximated by a linear function. It is shown that for moderately stable conditions ($0 < \frac{z-d}{L} < 1$) stability correction term $\Psi_m$ holds the following form:

$$\Psi_m\left(\frac{z-d}{L}\right) = -\beta * \left(\frac{z-d}{L}\right) \qquad\qquad \text{(eq. A)}$$

where $\beta$ is a constant. We consider a simple linear regression with offset parameter $a$ and slope parameter $b$. If we assume that $\Psi_m$ is linear, we can rewrite Eq. 1 in the following form:

$$\frac{k\,u(z)}{u^*} = a + b\left(\frac{z-d}{L}\right) \qquad\qquad \text{(eq. A)}$$

Now $a$ provides an estimate of $\ln(z-d)/z_0$ , and we can directly compute $z_0$ from $(z-d)/\exp(a)$. We use observations from 2014 only, unless stated otherwise in Table 1. For forest we assume that $d = (2/3) * h$ (Maurer et al., 2013), and we use the forest canopy height derived from GLAS. For short vegetation we assume that displacement height $d$ is negligible, that is, $d = 0$ . Graf et al., 2014 illustrated that the linearity approximation of $\Psi_m$ is valid for small negative values of $(z-d)/L$ , so we first select all points where $-0.1 < (z-d)/L < 1$. We filter out observation during rainfall and where $u^* < 0.15$, as presented in Chen et al., 2015. We split our data into a group with stable conditions ($L > 0$) and with unstable conditions ($L < 0$). We assume that the $z_0$ is more or less constant over a period of 5 days. For each 5-day period we plot $ku(z)/u^*$ against $(z-d)/L$ and fit a simple line function using linear least-squares. The $z_0$ values are then computed from offset parameter $a$. We compute the mean, median, standard deviation and the range of the all computed $z_0$ values in one year. If the computed $z_0$ values for

stable and unstable conditions in one 5-day period differ more than 50% from their arithmetic mean they are filtered out.

## 4. Results

### 4.1. Comparison of the default and updated $z_0$ values

We used the CORINE/Smiatek land use map to combine all the updated $z_0$ values into one single map. The resulting composite map has a horizontal resolution of 500 by 500 metres and is shown in Figure 2.

The mean relative difference (MRD) between the default and updated $z_0$ values is presented in Figure 3. The largest positive differences occur in forested areas, meaning that the default $z_0$ values are lower than the updated $z_0$ values. The largest negative deviations occur in urban areas and areas with "grass". The updated $z_0$ values are generally

lower in the Netherlands and Belgium, and mostly higher in Germany. Table 4 gives an overview of the default $z_0$ values in LOTOS-EUROS and the mean and standard deviation of the new $z_0$ values for each of the DEPAC land use classes. The updated $z_0$ values for "arable land", "coniferous forest", "deciduous forest" and "other" are on average higher than the default $z_0$ values in LOTOS-EUROS. The updated $z_0$ values for "grass", "permanent crops" and "urban" are on average lower than the default $z_0$ values in LOTOS-EUROS.

### 4.2. Comparison to $z_0$ values from other studies


We compared the updated $z_0$ values to $z_0$ values from several studies (Wieringa, 1993;Silva et al., 2007;Troen and Petersen, 1989;Lankreijer et al., 1993;Yang and Friedl, 2003), and $z_0$ values used in other CTMs (Simpson et al., 2012;Mailler et al., 2017). Table 5 gives an overview of these $z_0$ values. There is in general good agreement with these $z_0$ values, and the updated $z_0$ values mostly fall within the stated ranges. The updated $z_0$ values for coniferous

and deciduous forest are on the high side compared to these studies. A histogram of the forest canopy heights derived from GLAS within our study area is given in Figure S1. These differences can in part be explained by the occurrence of relatively tall forest canopy (~30 meters) in the dataset, especially in forest in southern Germany, whereas most of these studies either assumed or studied shorter trees. Another possible explanation lies in the fact that we used a relatively large conversion factor of 0.136 (eq. 4), whereas a factor of 0.10 is also used quite often.

### 4.3. Comparison to $z_0$ values derived from EC measurement sites


We computed the $z_0$ values of the EC sites. We compared the $z_0$ values based on their land use stated by FLUXNET, to avoid issues arising from discrepancies in land use classifications. The forest sites (DBF, ENF and MF) are compared to $z_0$ values derived from GLAS. The cropland and wetland sites (CRO and WET) are compared to the NDVI-dependent $z_0$ values derived using the mean function shown in Figure 1. The grassland sites (GRA) are

compared to the NDVI-dependent $z_0$ values for grassland specifically. The results per site are given in Table 6. Figure 4 shows the comparison of the $z_0$ values from EC measurements and the updated $z_0$ values for different land use classes. The $z_0$ values for forest match quite well. The $z_0$ values for short vegetation seem to be overestimated for crops and underestimated for grassland and wetland sites. The underestimation of some grassland and wetland

sites can be explained by the large inter-site differences in vegetation cover. Some of the FLUXNET sites classified

as grasslands are for instance mostly covered with short grass only (for instance Oensingen), whereas there are also sites with relatively tall herbaceous vegetation, such as reeds (for instance Horstermeer). Compared to the default $z_0$ values in LOTOS-EUROS, the root-mean-square difference (RMSD) improved from 0.76 (default $z_0$) to 0.60 (updated $z_0$). For forest, the RMSD decreased from 1.23 (default $z_0$) to 0.96 (updated $z_0$). For short vegetation, the RMSD also decreased, from 0.22 (default $z_0$) to 0.19 (updated $z_0$). Figure 5 shows the comparison of the seasonal

variation in computed and satellite-derived $z_0$ values for the FLUXNET sites classified as crops in 2014. We can once more observe a clear offset between the two. The FLUXNET $z_0$ values go to near-zero values in wintertime, whereas the satellite-derived $z_0$ values never drop below 0.12 meters. This seems to be due to the distribution of the NDVI values (Figure S1), which shows that the NDVI $> 0.4$ most of the time. The seasonal patterns, on the other hand, seem to match well in general, even though the satellite-derived $z_0$ values rise somewhat earlier in the year.

**4.4. Comparison of the default and MODIS LAI**

The yearly mean MODIS LAI values are shown in Figure 6. The mean differences between the MODIS and the default LAI values are presented in Figure 7. The largest positive differences occur in areas with "arable land", where the mean default LAI values are lower than the MODIS LAI values. The largest negative deviations occur in areas with forest, especially for "coniferous forest". The seasonal variation of the MODIS and the default LAI

values are shown in Figure 8. The default LAI of "grass" and "deciduous forest" seems to fit the yearly variation of the MODIS LAI quite well. We matched the MODIS-LAI with the locations of the FLUXNET sites to take a closer look at the pattern for different land use classes. Figure 9 shows the seasonal variation of the MODIS-LAI at FLUXNET sites with different land use classifications. The LAI of the grassland sites seems to vary the most, which corresponds to the large inter-site differences in vegetation cover. For the cropland sites, we can recognize the

growing season and the apparent harvest, where the LAI values drop again. Of all the different land use classes, deciduous broadleaf forest sites reach the highest LAI values in the growing season. There is less variation in the LAI for evergreen needle leaf forest sites. However, the wintertime LAI values seem to be unrealistically low.

**4.5. Implications for modelled $N_r$ deposition fields**

In the following section, the impact of the updated LAI and $z_0$ values on modelled $N_r$ deposition fields in LOTOS-

EUROS is discussed. A total of four different LOTOS-EUROS runs are compared to examine the individual effect of the updated LAI and $z_0$ values on the modelled $N_r$ distributions and deposition fields. The first run, named "LE$_{default}$", is the standard run using default LAI and $z_0$ values. The second run, named "LE$_{LAI}$", uses updated LAI values and the default $z_0$ values. The third run, named "LE$_{z0}$", uses updated $z_0$ values and the default LAI values. The fourth run, named "LE$_{z0+LAI}$", uses both updated LAI and $z_0$ values. From now on, we will refer to the outputs of

these different runs using the abovementioned abbreviations.

**4.5.1.    Effect on total $N_r$ deposition**

Figure S3 shows the division of the total terrestrial $N_r$ deposition over Germany, the Netherlands and Belgium into different $N_r$ compounds for each of the model runs. A relatively larger portion of the total $N_r$ deposition is attributed to oxidized forms of $N_r$ in Germany. The reduced forms of $N_r$ predominate in the Netherlands and Belgium. The largest change in total $N_r$ deposition occurs in Belgium (+6.19%) due to the inclusion of the MODIS LAI. This corresponds to the relative increase in LAI values here. The inclusion of the updated $z_0$ values lead to a minor decrease in total $N_r$ deposition in the Netherlands (-1.45%) and Belgium (-1.13%), and a minor increase in Germany (+0.44%).

### 4.5.2. Effect on wet and dry $N_r$ deposition

We examined the direct effect of the updated LAI and $z_0$ values on the modelled dry $N_r$ deposition, as well as the related indirect effect in modelled wet $N_r$ deposition. Figure 10 shows the dry and wet $N_r$ deposition in kg N ha$^{-1}$ in 2014, modelled with the updated LAI and $z_0$ values as input in LOTOS-EUROS. Figure 11 shows the relative changes in the total amount of dry and wet $N_r$ deposition of the different runs with respect to the default run. The combined effect shows an increase of the amount of dry $N_r$ deposition over most parts of Belgium and Germany. The amount of wet $N_r$ deposition decreases over most parts of Germany and eastern Belgium, but remains unchanged in north-western parts of Germany. We observe a decrease in total $N_r$ deposition in the Netherlands. In general, we observe changes ranging from approximately -20% to +30% in the total amount of dry $N_r$ deposition. The changes in wet $N_r$ deposition are smaller in magnitude and range from approximately -3% to +3%.

### 4.5.3. Effect on reduced and oxidized $N_r$ deposition

We split up the total Nr deposition into $NH_x$ ($NH_3$ and $NH_4^+$) and $NO_y$ (NO and $NO_2$ and $NO_3^-$ and $HNO_3$) deposition, to look at the effect of the updated LAI and $z_0$ input maps on the deposition of reduced and oxidised forms of $N_r$, respectively. Figure 12 shows the modelled $NH_x$ and $NO_y$ deposition in kg N ha$^{-1}$ in 2014, including the updated LAI and $z_0$ input values. Figure 13 shows the relative changes (%) in the total $NH_x$ and $NO_y$ of the different runs with respect to the default run of LOTOS-EUROS. The updated $z_0$ values have a larger impact on the $NH_x$ deposition than on the $NO_y$ deposition. The implementation of the updated $z_0$ values has led to a decrease in $NH_x$ deposition over most of the Netherlands, and western Belgium, driven by the large fraction of grassland here. The updated LAI values led to relatively more $NH_x$ deposition in Belgium. The updated LAI values led to an increase of $NO_y$ deposition in almost all areas within the modelled region, except for some urban areas. Moreover, the impact seems to be limited in large forests located in background areas.

### 4.5.4. Effect on land use specific fluxes

Table 7 gives an overview of changes in the distribution of the land use specific fluxes in the whole study area (Germany, the Netherlands and Belgium combined) for the different runs. The most distinct changes in $N_r$ deposition are due to the updated LAI values ("$LE_{LAI}$"), where we observe an increase in total $N_r$ deposition on urban areas (+ 16.62%) and arable land (+ 9.53%), and a decrease on coniferous forests (- 9.36%). This coincides with the categories where we observe the largest changes in LAI values. The default LAI values in urban areas were

first set to zero for all urban DEPAC categories. The MODIS LAI values, however, are only zero in densely populated areas and areas with industry. The main effects of the updated $z_0$ values ("$LE_{z0}$") can be observed for grass (-3.95 %), permanent crops (+ 3.27) and arable land (-3.17 %). In the combined run, "$LE_{z0+LAI}$", we observe an amplified effect in total $N_r$ deposition over grass (- 8.05%) and arable land (+ 12.88%). The impact of the individual parameters on the remaining land use categories is attenuated in this run.

### 4.6. Implications for $N_r$ distributions

The changes in $N_r$ deposition amounts induce an effect in the modelled distribution of nitrogen components. Here, we look at the effect of the updated LAI and $z_0$ values on $NH_3$ and $NO_2$ surface concentrations. Figure 14 shows the updated modelled $NH_3$ and $NO_2$ surface concentrations in 2014. The dots on top of the figures represent the stations that are used for validation, and their observed mean $NH_3$ and $NO_2$ surface concentrations. Figure 15 shows the relative change in yearly mean $NH_3$ and $NO_2$ surface concentrations in 2014 of the different runs with respect to the default run of LOTOS-EUROS.

The first column represents the changes in $NH_3$ and $NO_2$ surface concentrations due to the updated $z_0$ values. The $NH_3$ surface concentration in the Netherlands has increased by up to ~8%. The $NH_3$ surface concentration in almost all of Germany has decreased by up to ~10%. The changes in the $NO_2$ surface concentration are less distinct and changed approximately minus to plus 4% in most areas. The middle column represents the changes in $NH_3$ and $NO_2$ surface concentrations due to the inclusion of the MODIS LAI only. The $NH_3$ surface concentration has increased with up to ~10% in the Netherlands, western Belgium and north-western and southern Germany. The $NH_3$ surface concentration has decreased in eastern Belgium and central Germany. The $NO_2$ surface concentrations have decreased with up to ~6% in almost all of the modelled area.

### 4.7. Comparison to in-situ measurements

To analyse the effect of the updated LAI and $z_0$ values, we compared our model output to the available in-situ observations. Due to the lack of available dry deposition measurements, we decided to use $NH_4^+$ and $NO_3^-$ wet deposition and $NH_3$ and $NO_2$ surface concentrations measurements instead. The distribution of the wet deposition stations is shown in Figure 16, as well as the modelled mean $NH_4^+$ (left) and $NO_3^-$ (right) wet deposition in 2014. The locations of the stations that measure the $NH_3$ and $NO_2$ surface concentrations are shown in Figure 14.

The relationships between the modelled and observed fields are evaluated using the Pearson's correlation coefficient (r), the RMSD and the coefficients (slope, intercept) of simple linear regression. Table S1 shows these measures for the comparison with monthly mean $NO_3^-$ wet deposition, $NH_4^+$ wet deposition, and the monthly mean $NH_3$ and $NO_2$ surface concentrations in 2014. Table S1 shows the same statistics but computed per DEPAC land use class. Overall, we do not observe large changes in the shown measures due to the inclusion of the updated LAI and $z_0$ values on a yearly basis. The model underestimates $NO_3^-$ wet deposition, and $NH_4^+$ to a lesser extent, too. The modelled $NH_3$ surface concentrations are in general higher than observed concentrations. The opposite is true for

NO$_2$, here, the modelled surface concentrations are lower than the observed concentrations. The computed measures
did not change drastically due to the inclusion of the updated z$_0$ and LAI values.

Figure S4 shows the monthly mean NO$_3^-$ wet deposition,  NH$_4^+$ wet deposition, NH$_3$ surface concentration and NO$_2$
surface concentrations for the different model runs and the mean of the corresponding in situ observations. The
relative changes with respect to the default model run are shown in the bottom figures. For NH$_4^+$, the model captures
the observed pattern quite well, although the mean spring peak has slightly shifted. The model captures the monthly
variation of NO$_3^-$ well in general, too. There appears to be an underestimation during the winter, especially in
December. The observed NH$_3$ surface concentrations are lower than the modelled concentrations at the beginning of
spring and higher during summer. The measured NO$_2$ surface concentrations are continuously higher than the
modelled values. A potential reason for this might be the spread of the NO$_2$ stations. Unlike NH$_3$, NO$_2$ is not only
measured in nature areas but all types of locations. Even the selected background stations may, therefore, be located
relatively closer to emission sources, leading to higher observed NO$_2$ surface concentrations. The changes due to the
inclusion of either the MODIS LAI or the updated z$_0$ values in our model are limited.

Both Table S1, Table S2 and Figure 16 illustrate that the comparability of the modelled wet deposition and surface
concentration fields to the available in-situ measurements did not change significantly. The impact of the updated
LAI and z$_0$ values on these fields is largely an indirect effect of the more distinct changes in the dry deposition, and
thus too small to lead to any drastic changes. We conclude that we are unable to demonstrate any major
improvements with the use of the currently available in-situ measurements.

## 5.    Discussion

This paper aimed to improve the surface characterization of LOTOS-EUROS through the inclusion of satellite-
derived leaf area index (LAI) and roughness length (z$_0$) values. We used empirical functions to derive roughness
length (z$_0$) values for different land use classes. The updated z$_0$ values are compared to literature values, showing a
good agreement in general. We also compared the z$_0$ values to z$_0$ values computed from FLUXNET sites. The z$_0$
values for forest seemed to match well. The z$_0$ values for short vegetation seem to be overestimated for crops and
underestimated for grassland and wetland sites. The differences for short vegetation types can be partially explained
by the large inter-site variability in vegetation types within each classified land use (e.g. reeds versus short grass).
The equation used for grassland in this study seems to work best for short grasslands. For our current study area, this
does not pose a problem, since most grasslands in Germany, Belgium and the Netherlands are managed and grazed
upon. We found an improved RMSD value of 0.60, compared to RMDS of 0.76 with default z$_0$ values. Even though
there is an offset between the satellite-derived and computed FLUXNET z$_0$ values for crops, the seasonal pattern
seemed to match well. The offset can be explained by the absence of low NDVI (<0.4) values.

The z$_0$ is closely related to the geometric features and distributions of the roughness elements in a certain area. The
updated z$_0$ values are linked to specific land use pixels and are therefore assumed reasonable estimates for
moderately homogeneous areas with this specific land use type. There are various approaches to combine these z$_0$
values into an 'effective' roughness for larger, mixed areas (e.g. (Claussen, 1990;Mason, 1988)). The LOTOS-

EUROS model uses logarithmic averaging to compute an effective roughness for an entire model pixel. This
averaging step seems to be one of the reasons why the effect of our updated $z_0$ values on the deposition fields is
limited. To illustrate this, the relative change in total dry $NH_3$ deposition due to the updated $z_0$ values were
computed and shown in Figure 17. We used increasing threshold percentages to sort the $NH_3$ deposition on a model
pixel level per land use type and fraction. Figure 17 shows that the differences in total $NH_3$ deposition between the
two runs increase with increasing land use fraction. The model pixels that mostly consist of one type of land use
seem to show the largest change in $NH_3$ deposition. The change thus appears to be less distinct in pixels that have a
higher degree of mixing. Most of the model pixels largely contain mixtures of different land uses on the current
model resolution. As a result, averaging of $z_0$ on a model pixel level is thus likely to cause a levelling effect on the
current model resolution. The impact of the updated $z_0$ values is therefore expected to be larger at a higher model
resolution. The use of another approach for computing the 'effective' roughness could potentially lead to stronger
changes in the modelled deposition fields.

Moreover, we should also consider the limitations of the datasets used in this study. The forest canopy height map
used in this study has been validated against 66 FLUXNET sites (Simard et al., 2011). The results showed that
RMSE = 4.4 m and $R^2$ = 0.7 after removal of 7 outliers. For the FLUXNET forest sites used in this study, we
compared the forest canopy heights from GLAS to the maximum forest canopy height at the FLUXNET sites taken
from (Flechard et al., 2019). For all but one site (DE-Hai), the forest canopy heights from GLAS were lower than
this value (Table S3). This method could potentially be improved by using another product with either a higher
precision or resolution. For modelling studies on a national level one could or instance consider the use of airborne
LiDAR point clouds to retrieve forest canopy heights. This procedure, although it is computationally expensive,
would allow us to create high resolution $z_0$ maps.

The MODIS-LAI at the FLUXNET sites showed realistic seasonal variations for most land use classifications,
except for relatively low winter-time values for evergreen needle leaf forests. The previous versions of the MODIS-
LAI have been validated in many studies (e.g. (Fang et al., 2012;Wang et al., 2004;Kobayashi et al., 2010)),
showing an overall good agreement with ground observed LAI values and other LAI products. The seasonality in
LAI is properly captured for most biomes, but unrealistic temporal variability is observed for forest due to
infrequent observations. Also, the previous versions overestimate LAI for forests (Fang et al., 2012;Kobayashi et al.,
2010;Wang et al., 2004). The MODIS-LAI products have been gradually improving with each update, however,
these issues still exist in the newer versions of the product. For the most recent version of the MODIS-LAI, version
6, Yan et al. (2016) found an overall RMSE of 0.66 and a $R^2$ of 0.77 in comparison with LAI ground observation.
More recently, using a different approach, Xu et al. (2018) found a slightly higher RMSE of 0.93 and a $R^2$ of 0.77.
Some studies (e.g. Tian et al., 2004) have reported an underestimation of the MODIS-LAI in presence of snow-
cover, particularly affecting evergreen forests. There was only a limited amount of snowfall in our study region, so
this did not lead to any problems. However, this issue should be carefully considered when using the MODIS-LAI
for regions with frequent snow-cover, like Scandinavia. LOTOS-EUROS uses meteorological data to determine
what regions are covered with snow. If a certain location is snow-covered, the standard parameterization for the

canopy resistance is not used. LOTOS-EUROS uses a pre-defined value for the canopy resistance instead. As such, these low MODIS-LAI values do not affect the modelled deposition in LOTOS-EUROS during snow-cover.

Though the issues with the MODIS-LAI should be considered with care, the spatial and temporal distribution of these LAI values is more realistic than that of the default LAI values used in LOTOS-EUROS. The same holds for the updated, time-variant $z_0$ values. The representation of the growing season is now more realistic due to their

dependence on NDVI and LAI values. Moreover, the $z_0$ values for forests now also have a clear spatial variation, such as a latitudinal gradient with increasing $z_0$ values towards the south of Germany. These type of patterns can simply not be captured by fixed values.

We evaluated the effect of updated $z_0$ and LAI values on modelled $N_r$ distribution and deposition fields. The distribution of the relative changes in deposition of the reduced and oxidized forms of reactive nitrogen showed a

similar pattern. Here, the updated $z_0$ values led to a variation of ~±8%, and the updated LAI values led to variations of ~±30% in both fields. The dry deposition fields were most sensitive to changes in $z_0$ and LAI, as these varied from approximately -20% to +20% with the updated $z_0$ map, and from -20% even up to +30% with the MODIS LAI values. As a result, we observed a shift from wet to dry deposition, except for the Netherlands, where we observe an opposite shift, from dry to wet deposition. Moreover, we observed a redistribution of $N_r$ deposition over different

land use classes on a sub grid level. To illustrate the potential consequences on a local scale, we computed the critical load exceedances for deciduous and coniferous forest (Figure 18) using critical loads of 10 kg following Bobbink et al. (2010b). Compared to the default run, the changes may be sizable locally, ranging from approximately -3 kg up to +2 kg for deciduous forest and even over -3 kg for coniferous forest.

The uncertainties of the LAI and $z_0$ input data are but one aspect of the model uncertainty of CTMs. The model

uncertainty has several other origins, like the physical parameterizations (e.g. deposition velocities) and the numerical approximations (e.g. grid size). Two of the most important uncertainties related to deposition modelling are the emissions and the surface exchange parameterization. The emission inventories for reactive nitrogen hold a relatively large uncertainty. The uncertainty of the reported annual total $NH_3$ emissions is estimated to be at least ±30%. This is mainly due to the diverse nature of agricultural emission sources, leading to large spatio-temporal

variations. The annual $NO_x$ emissions total hold a lower uncertainty, of approximately ±20% (Kuenen et al., 2014). Emissions at specific locations, especially for $NH_3$, are even more uncertain due to assumptions made in the redistribution and timing of emissions. A recent paper of Dammers et al., 2019, for instance, found that satellite-derived $NH_3$ emissions of large point sources are a factor 2.5 higher than those given in emission N-inventories. The surface-exchange parameterization is another source of uncertainty. The complexity of the $NH_3$ surface

exchange schemes in CTMs is usually low compared to the current level of process understanding (Flechard et al., 2013). Moreover, large discrepancies exist between deposition schemes. Flechard et al. (2011) for instance showed that the differences between four dry deposition schemes for reactive nitrogen can be as large as a factor 2-3.

This work has shown that changes in two of the main deposition parameters (LAI, $z_0$) can already lead to distinct, systematic changes (~30%) in the modelled deposition fields. This demonstrates the model's sensitivity toward these

input values, especially the LAI. In addition to the known uncertainty involved with surface exchange parameterization itself, this further stresses the need for further research. Another important aspect that should receive more attention is the validation of the dry deposition fields with in-situ dry deposition measurements. Here we illustrated the need for direct validation methods, as relatively large changes in modelled dry deposition field cannot be verified sufficiently by surface concentration and wet deposition measurements.

The surface-atmosphere exchange remains one of the most important uncertainties in deposition modelling. The use of satellite products to derive LAI and $z_0$ values can help us to represent the surface characterization in models more accurately, which in turn might help us to minimize the uncertainty in deposition modelling. The approach to derive high resolution, dynamic $z_0$ estimates presented here can be linked to any land use map and is as such transferable to many different models and geographical areas.

**Data availability.** LOTOS-EUROS v2.0 is available for download under license at https://lotos-euros.tno.nl/. The adjusted model routines to read in the $z_0$ and LAI values are available in cooperation with TNO and can be disclosed upon request. All MODIS products used in this study are open-source and can be downloaded at https://ladsweb.nascom.nasa.gov. The global forest canopy height dataset can be downloaded at https://daac.ornl.gov/. The population density grid can be downloaded at https://www.eea.europa.eu/.

**Acknowledgements.** We would like to thank the Umweltbundesamt (UBA), the German monitoring networks, as well as the European Monitoring and Evaluation Programme (EMEP) and the Rijksinstituut voor Volksgezondheid en Milieu (RIVM) for providing the in-situ observations used for validation. This work used eddy covariance data acquired and shared by the FLUXNET community, including these networks: AmeriFlux, AfriFlux, AsiaFlux, CarboAfrica, CarboEuropeIP, CarboItaly, CarboMont, ChinaFlux, Fluxnet-Canada, GreenGrass, ICOS, KoFlux,
LBA, NECC, OzFlux-TERN, TCOS-Siberia, and USCCC. The ERA-Interim reanalysis data are provided by ECMWF and processed by LSCE. The FLUXNET eddy covariance data processing and harmonization was carried out by the European Fluxes Database Cluster, AmeriFlux Management Project, and Fluxdata project of FLUXNET, with the support of CDIAC and ICOS Ecosystem Thematic Center, and the OzFlux, ChinaFlux and AsiaFlux offices.

**Author Contribution.** JWE, MS and SG designed the research. RK, AS and SG performed the model simulations. SG did the input data processing and model output analysis. All authors contributed to the interpretation of the results. SG wrote the paper with contributions from all co-authors.

**Competing interests.** The authors declare that they have no conflict of interest.

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

Table 1: Overview of eddy-covariance sites used to compute $z_0$. The following abbreviations for land use types are used: DBF - Deciduous broadleaf forest, ENF - Evergreen needle leaf coniferous forest, MF - Mixed forest, CRO – Croplands, WET – Permanent wetlands, GRA – Grasslands

| Site ID | Site name | Latitude | Longitude | Land use | z (m) | FLUXNET2015 DOI |
| --- | --- | --- | --- | --- | --- | --- |
| BE-Bra | Brasschaat | 51.30761 | 4.51984 | MF | 41.0 | 10.18140/FLX/1440128 |
| BE-Lon | Lonzee | 50.5516 | 4.74613 | CRO | 2.7 | 10.18140/FLX/1440129 |
| BE-Vie | Vielsalm | 50.30496 | 5.99808 | MF | 51.0 | 10.18140/FLX/1440130 |
| DE-Akm | Anklam | 53.86617 | 13.68342 | WET | 9.5 | 10.18140/FLX/1440213 |
| DE-Geb | Gebesee | 51.1001 | 10.9143 | CRO | 3.5 | 10.18140/FLX/1440146 |
| DE-Gri | Grillenburg | 50.95004 | 13.51259 | GRA | 3.0 | 10.18140/FLX/1440147 |
| DE-Hai (2012) | Hainich | 51.07917 | 10.4530 | DBF | 42.0 | 10.18140/FLX/1440148 |
| DE-Kli | Klingenberg | 50.89306 | 13.52238 | CRO | 3.5 | 10.18140/FLX/1440149 |
| DE-Obe | Oberbärenburg | 50.78666 | 13.72129 | ENF | 30.0 | 10.18140/FLX/1440151 |
| DE-RuR | Rollesbroich | 50.62191 | 6.30413 | GRA | 2.6 | 10.18140/FLX/1440215 |
| DE-RuS | Selhausen Juelich | 50.86591 | 6.44717 | CRO | 2.5 | 10.18140/FLX/1440216 |
| DE-Seh (2010) | Selhausen | 50.87062 | 6.44965 | CRO | 2.0 | 10.18140/FLX/1440217 |
| DE-SfN | Schechenfilz Nord | 47.80639 | 11.3275 | WET | 6.7 | 10.18140/FLX/1440219 |
| DE-Tha | Tharandt | 50.96235 | 13.56516 | ENF | 42.0 | 10.18140/FLX/1440152 |
| NL-Hor (2011) | Horstermeer | 52.24035 | 5.0713 | GRA | 4.3 | 10.18140/FLX/1440177 |
| NL-Loo (2013) | Loobos | 52.16658 | 5.74356 | ENF | 26 | 10.18140/FLX/1440178 |
| FR-Fon | Fontainebleau-Barbeau | 48.47636 | 2.7801 | DBF | 37 | 10.18140/FLX/1440161 |
| CH-Cha | Chamau | 47.21022 | 8.41044 | GRA | 2.4 | 10.18140/FLX/1440131 |
| CH-Fru | Früebüel | 47.11583 | 8.53778 | GRA | 2.5 | 10.18140/FLX/1440133 |
| CH-Lae | Laegern | 47.47808 | 8.3650 | MF | 47 | 10.18140/FLX/1440134 |
| CH-Oe1 (2008) | Oensingen grassland | 47.28583 | 7.73194 | GRA | 1.2 | 10.18140/FLX/1440135 |
| CH-Oe2 | Oensingen crop | 47.28631 | 7.73433 | CRO | 2.25 | 10.18140/FLX/1440136 |
| CZ-wet | Trebon | 49.02465 | 14.77035 | WET | 2.6 | 10.18140/FLX/1440145 |


Table 2: An overview of the datasets that are used to derive $z_0$ input values for each DEPAC land use category.

| DEPAC class | Dataset |
| --- | --- |
| 1 - Grass | MODIS NDVI |
| 2 - Arable land | MODIS NDVI |
| 3 - Permanent crops | MODIS NDVI |
| 4 - Coniferous forest | GLAS forest canopy height |
| 5 - Deciduous forest | GLAS forest canopy height, MODIS LAI |
| 6 - Water | - |
| 7 - Urban | Population density |
| 8 - Other | MODIS NDVI |
| 9 - Desert | - |


Table 3: Studies that relate the aerodynamic roughness length ($z_0$) to satellite-derived NDVI values for specific land cover types and conditions.

| Function | | Vegetation type(s) | Reference |
| --- | --- | --- | --- |
| $z_0 = \begin{cases} -0.173 + 1.168 * NDVI - 1.125 * NDVI^2 & \text{for NDVI} < 0.6 \\ 0.1 & \text{for NDVI} \geq 0.6 \end{cases}$ | (eq. 6) | Partial cover cotton and wheat canopies | Hatfield (1988) |
| $z_0 = e^{-5.2+5.3*NDVI}$ | (eq. 7) | Alfalfa (Arizona) | Moran (1990) |
| $z_0 = e^{-5.5+5.8*NDVI}$ | (eq. 8) | Mixed, non-irrigated agricultural land (Mediterranean) | Bolle and Streckenbach (1993) |
| $z_0 = 0.2255 * NDVI + 0.0087$ | (eq. 9) | Spring maize | Yu et al. (2016) |
| $z_0 = 0.2476 * NDVI + 0.0615$ | (eq. 10) | Winter wheat | Yu et al. (2016) |
| $z_0 = 0.2858 * NDVI + 0.1017$ | (eq. 11) | Summer maize | Yu et al. (2016) |
| $z_0 = 0.0203 * NDVI^{0.9547}$ | (eq. 12) | Grassland | Xing et al. (2017) |


Table 4: An overview of the default and the adjusted roughness length of each DEPAC class used in LOTOS-EUROS for Germany, the Netherlands and Belgium. The datasets that are used to derive the updated $z_0$ values are given in the last column. The mean $z_0$ value of the DEPAC classes computed using the MODIS NDVI product is the yearly mean of all monthly $z_0$ values.

| DEPAC class | Default $z_0$ | New mean $z_0$ ($\sigma$) |
| --- | --- | --- |
| 1 - Grass | 0.03 | 0.013 (0.002) |
| 2 - Arable land | 0.1 | 0.19 (0.04) |
| 3 - Permanent crops | 0.25 | 0.20 (0.04) |
| 4 - Coniferous forest | 2 | 2.91 (0.56) |
| 5 - Deciduous forest | 2 | 2.55 (0.51) |
| 6 - Water | 0.002 | - |
| 7 - Urban | 2 | 0.92 (0.20) |
| 8 - Other | 0.05 | 0.19 (0.05) |
| 9 - Desert | 0.013 | - |




Table 5: Roughness length values from different types of studies. The first column states the global land use category of the $z_0$ values. The second column states the (range of) $z_0$ values, as well as the specific type of land use they are derived for. The third column shows the reference ([1]Literature study, [2]Model input (EMEP MSC-W), [3]Model input (CHIMERE)).

| Land use categories | Proposed $z_0$ values (type name) | Reference |
|---|---|---|
| Grass | 0.008 – 0.03 (short grass, moss) | Wieringa (1993) [1] |
| | 0.02 – 0.06 (long grass, heather) | |
| | 0.03 – 0.6 (categories DEPAC 1) | Silva et al. (2007) |
| | 0.022 (short grassland) | Gallagher et al. (2002) |
| | 0.063 (long grassland) | |
| | 0.01    (mown grass) | Troen and Petersen (1989) |
| | 0.03 | Simpson et al. (2012) [2] |
| | 0.1 (grassland) | Mailler et al. (2017) [3] |
| | 0.013 (mean DEPAC 1) | This study |
| Arable land / permanent crops | 0.04 – 0.09 (low mature crops) | Wieringa (1993) [1] |
| | 0.12 – 0.18 (high mature crops) | |
| | 0.05 – 0.5 (categories DEPAC 2) | Silva et al. (2007) |
| | 0.1 – 0.5 (categories DEPAC 3) | |
| | 0.12 (arable crop) | Gallagher et al. (2002) |
| | 0.03 - 0.1  (farmland) | Troen and Petersen (1989) |
| | 0.1 – 0.2 (different types of crops) | Simpson et al. (2012) [2] |
| | 0.05 – 0.15 (agriculture) | Mailler et al. (2017) [3] |
| | 0.19 (mean DEPAC 2) | This study |
| | 0.20 (mean DEPAC 3) | |
| Forests | 0.8 – 1.6 (mature pine forests) | Wieringa (1993) [1] |
| | 0.6 – 1.2 (categories DEPAC 4 & 5) | Silva et al. (2007) |
| | 1 | Troen and Petersen (1989) |
| | 0.8 – 1 | Simpson et al. (2012) [2] |
| | 1 | Mailler et al. (2017) [3] |
| | 2.5 | Gallagher et al. (2002) |
| | 1.71 – 1.9 (oak and pine trees) | Lankreijer et al. (1993) |
| | 1.70 – 2.29 (spruce and pine trees) | Yang and Friedl (2003) |
| | 2.91 (mean DEPAC 4) | |
| | 2.55 (mean DEPAC 5) | This study |
| Urban | 0.4 – 0.7 (dense low buildings) | Wieringa (1993) [1] |
| | 0.7 – 1.5 (regularly-built town) | |
| | 0.005 – 1.3 (categories DEPAC 7) | Silva et al. (2007) |
| | 0.5 – 1 (suburbs, city) | Troen and Petersen (1989) |
| | 1 | Simpson et al. (2012) [2] |
| | 1 | Mailler et al. (2017) [3] |
| | 0.92 (mean DEPAC 7) | This study |
| Other | 0.35 - 0.45 (continuous bushland) | Wieringa (1993) [1] |
| | 0.03 – 0.1 (categories DEPAC 8) | Silva et al. (2007) [1] |
| | 0.01 (heathland) | Gallagher et al. (2002) |
| | 0.05 – 0.2 (moorland, scrubs, wetlands) | Simpson et al. (2012) [2] |
| | 0.15 (scrubs) | Mailler et al. (2017) [3] |
| | 0.19 (mean DEPAC 8) | This study |


Table 6: Comparison of the computed $z_0$ values from FLUXNET observations and the corresponding satellite derived $z_0$ values. The table shows the mean and standard deviation of all $z_0$ values in one year. For forest, only the maximum $z_0$ value is given.

| Site ID | Land use | Computed $z_0$ | Satellite-derived $z_0$ |
|---------|----------|----------------|--------------------------|
| DE-Hai | DBF | 2.3 (0.88) | 3.5 |
| FR-Fon | DBF | 2.9 (0.53) | 2.6 |
| DE-Obe | ENF | 1.9 (0.52) | 3.4 |
| DE-Tha | ENF | 2.6 (0.72) | 3.1 |
| NL-Loo | ENF | 2.8 (0.43) | 2.3 |
| BE-Bra | MF | 2.8 (0.69) | 2.4 |
| BE-Vie | MF | 4.7 (1.5) | 3.0 |
| CH-Lae | MF | 3.5 (0.92) | 3.8 |
| DE-Akm | WET | 0.72 (0.21) | 0.14 (0.033) |
| DE-SfN | WET | 0.46 (0.13) | 0.22 (0.033) |
| CZ-wet | WET | 0.16 (0.075) | 0.18 (0.040) |
| BE-Lon | CRO | 0.013 (0.018) | 0.16 (0.029) |
| DE-Geb | CRO | 0.020 (0.020) | 0.17 (0.025) |
| DE-Kli | CRO | 0.043 (0.043) | 0.18 (0.030) |
| DE-RuS | CRO | 0.066 (0.048) | 0.15 (0.015) |
| DE-Seh | CRO | 0.034 (0.028) | 0.16 (0.015) |
| CH-Oe2 | CRO | 0.084 (0.065) | 0.19 (0.019) |
| DE-RuR | GRA | 0.035 (0.019) | 0.014 (0.0016) |
| NL-Hor | GRA | 0.14 (0.076) | 0.014 (0.0019) |
| CH-Cha | GRA | 0.078 (0.038) | 0.015 (0.00099) |
| CH-Fru | GRA | 0.13 (0.086) | 0.015 (0.0024) |
| CH-Oe1 | GRA | 0.0064 (0.0035) | 0.014 (0.00088) |


Table 7: Relative change (%) in total $N_r$ deposition w.r.t. the default run over Germany, the Netherlands and Belgium in 2014 per land use class.

| | $LE_{LAI}$ | $LE_{z0}$ | $LE_{z0+LAI}$ |
|---|---|---|---|
| Grass (1) | -5.34 | -3.95 | -8.05 |
| Arable land (2) | 9.53 | 3.27 | 12.88 |
| Permanent crops (3) | 0.22 | -3.17 | -2.78 |
| Coniferous forest (4) | -9.36 | -0.86 | -7.36 |
| Deciduous forest (5) | 1.15 | 1.93 | 1.83 |
| Urban (7) | 16.62 | -0.37 | 6.53 |
| Other (6,8,9) | 3.45 | 0.58 | 3.31 |

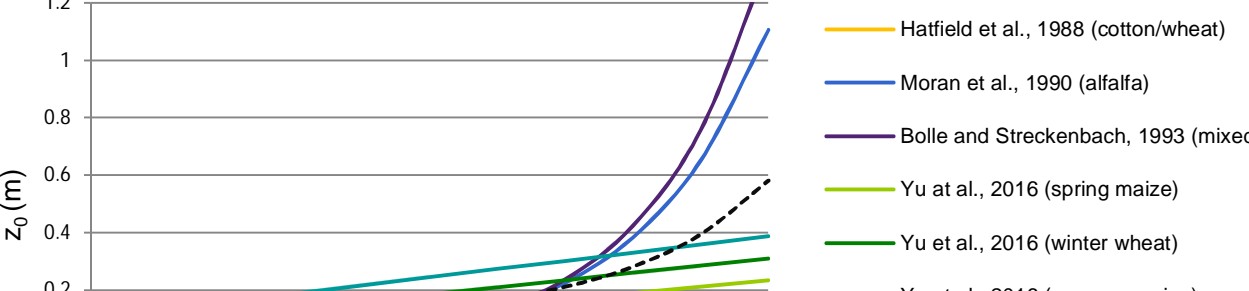

Figure 1: Several functions that relate the roughness length, $z_0$ (m) to the normalized difference vegetation index (NDVI). The dotted line shows the average of all the functions. This function is used to compute NDVI-dependent $z_0$ values for the subcategories within DEPAC classes "arable", "other" and "permanent crops".

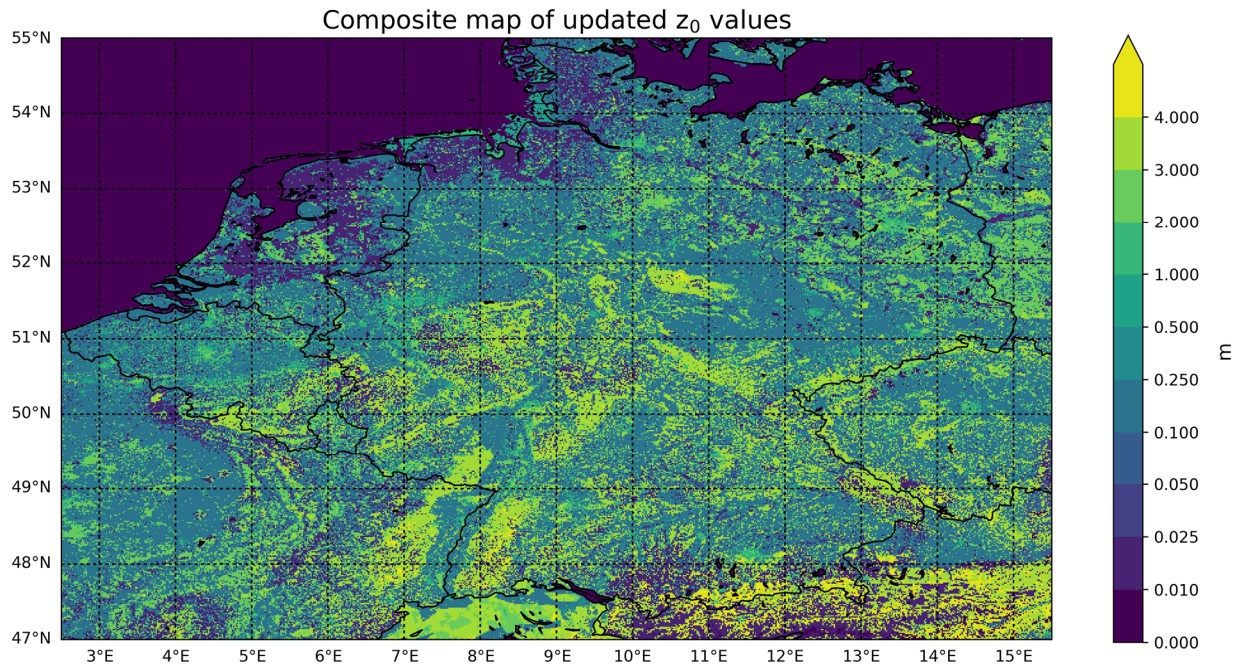

Figure 2: Composite map of the new $z_0$ values. The yearly mean is displayed for land use classes with time-variant $z_0$ values.

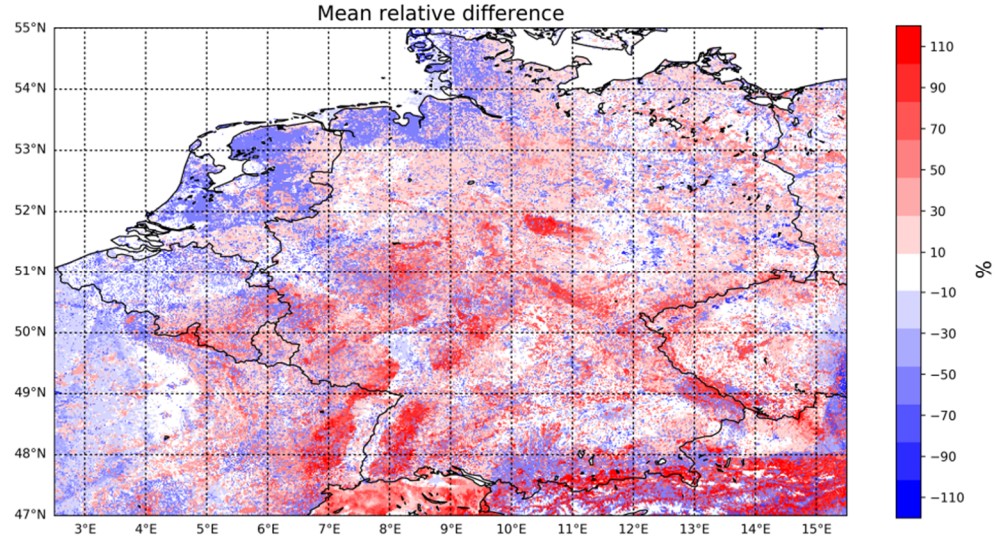

Figure 3: Mean relative difference (%) of the updated $z_0$ values with respect to the default $z_0$ values.

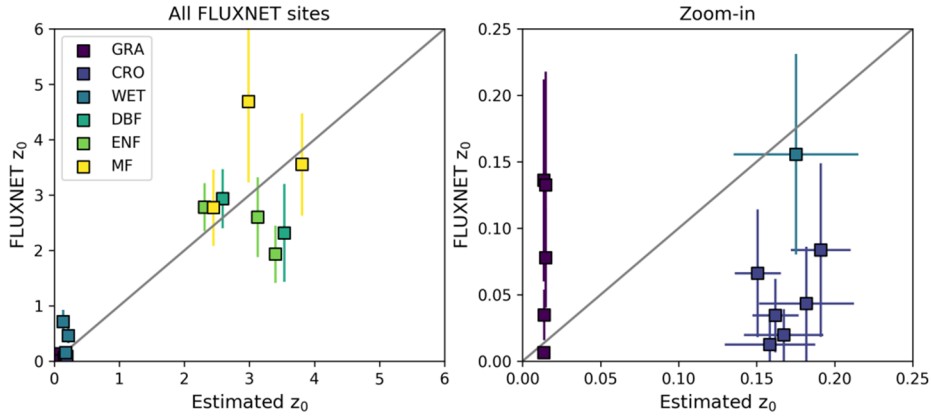

Figure 4: Comparison of the updated $z_0$ values (x-axis) to the $z_0$ values derived from EC measurements (y-axis). The error bars
indicate the standard deviation.

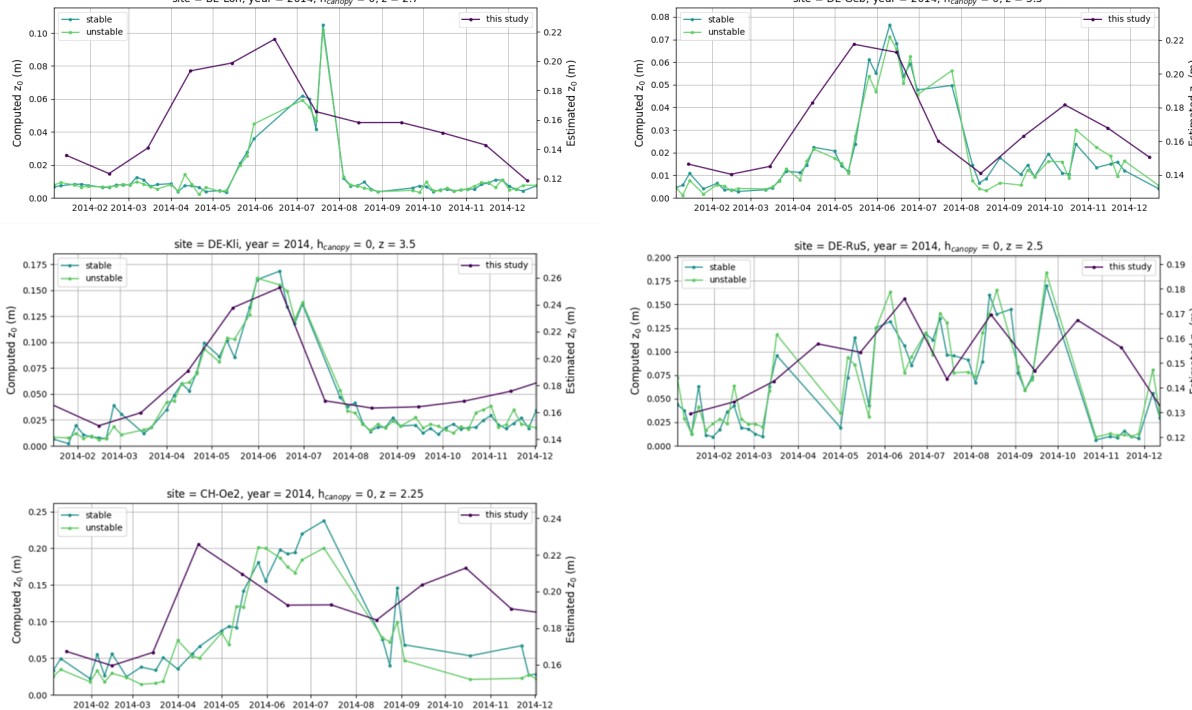

Figure 5: Seasonal variation of the computed $z_0$ values at the FLUXNET cropland sites in 2014 (left-axis), and the corresponding $z_0$ values estimated from NDVI-values (right-axis). The assumptions used to compute the $z_0$ values are shown in the titles of the figures.

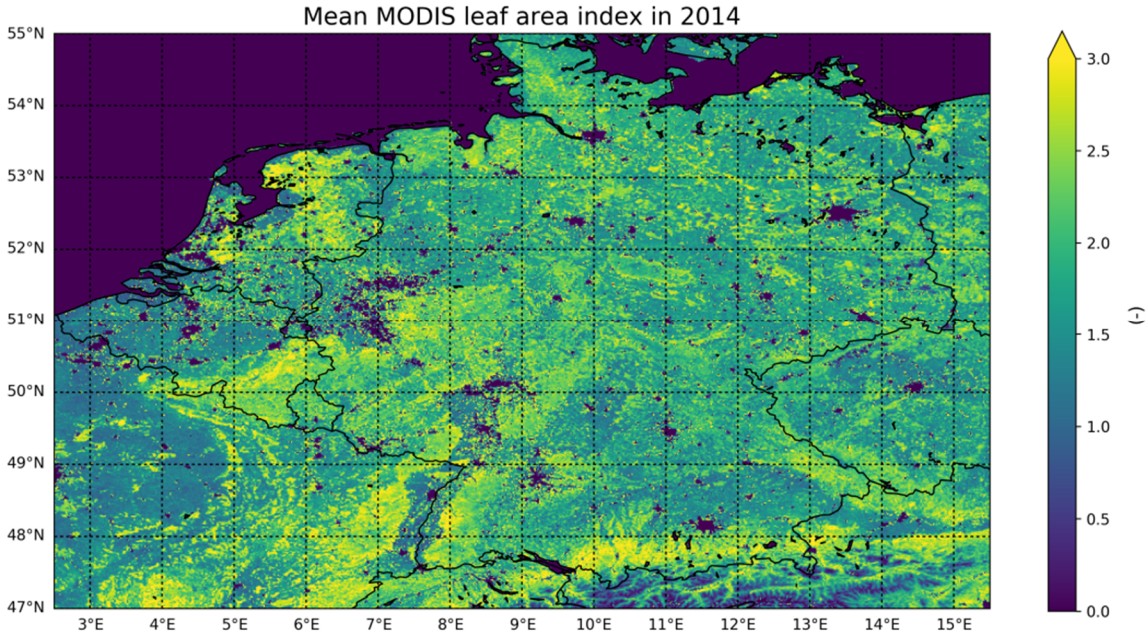

Figure 6: The yearly mean MODIS leaf area index in 2014.

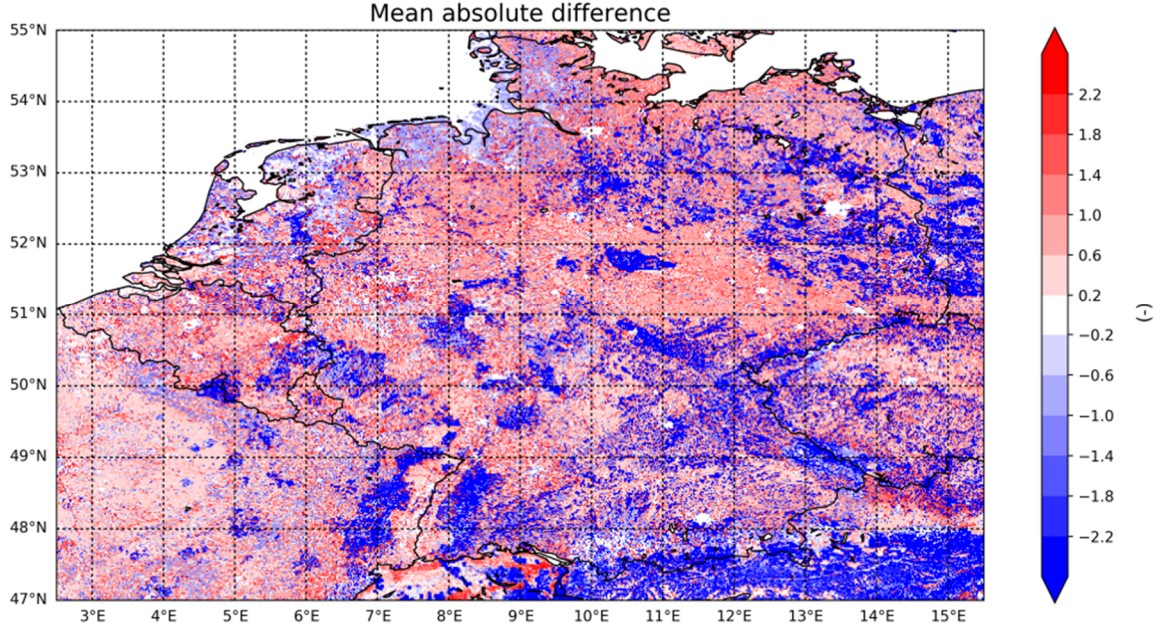


Figure 7: The absolute differences ($LAI_{MODIS} - LAI_{default}$) between the MODIS leaf area index and the default leaf area index in LOTOS-EUROS.

Leaf area index of DEPAC classes in 2014

Figure 8: Seasonal variation of the default and MODIS-LAI values per DEPAC class. The black line represents the mean

MODIS-LAI of all pixels within the modelled grid for that particular DEPAC class, the ranges represent the mean plus and minus the standard deviation of the MODIS-LAI. The red line depicts the default LAI values in LOTOS-EUROS.

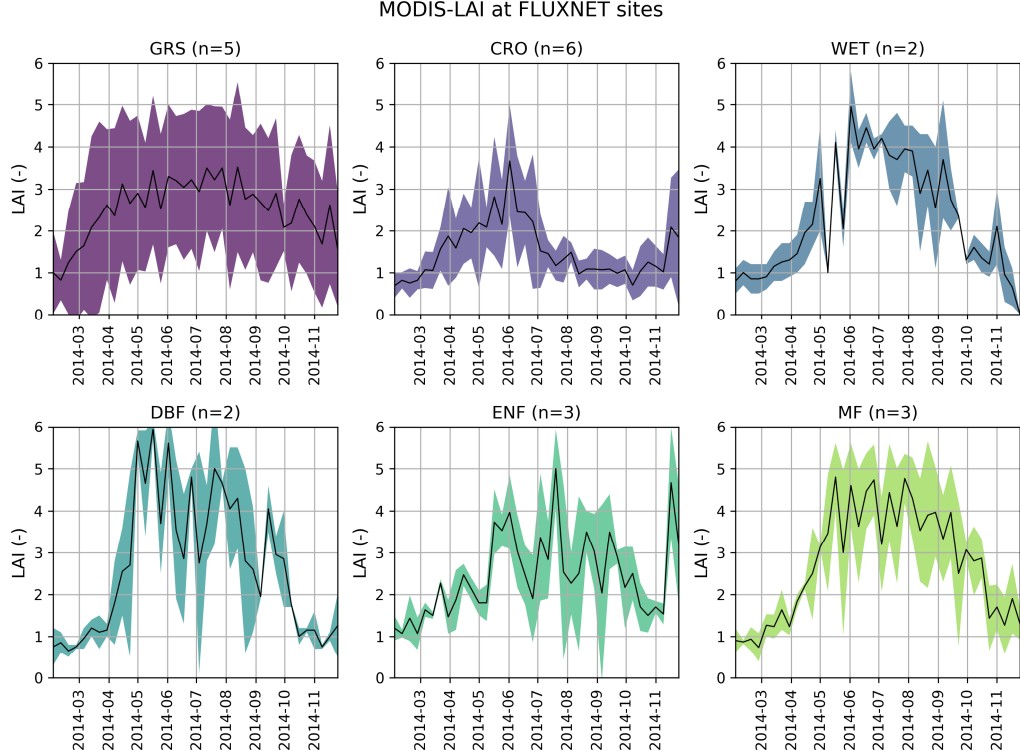

Figure 9: Seasonal variation of the MODIS-LAI at FLUXNET sites with different land use classifications. The black line represents the mean MODIS-LAI value per land use and the ranges represent the mean plus and minus the standard deviation.



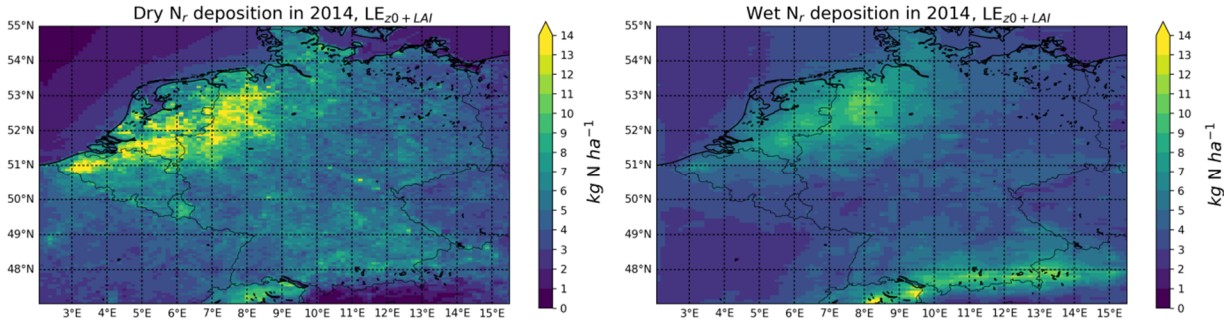

Figure 10: The modelled amount of dry (left) and wet (right) deposition in kg N ha⁻¹ in 2014.

Relative change (%) in total dry $N_r$ deposition in 2014

Relative change (%) in total wet $N_r$ deposition in 2014

$(LE_{z0} - LE_{default})/LE_{default} * 100$

$(LE_{z0} - LE_{default})/LE_{default} * 100$

$(LE_{LAI} - LE_{default})/LE_{default} * 100$

$(LE_{LAI} - LE_{default})/LE_{default} * 100$

$(LE_{z0+LAI} - LE_{default})/LE_{default} * 100$

$(LE_{z0+LAI} - LE_{default})/LE_{default} * 100$


Figure 11: The relative change in total dry (left) and wet (right) $N_r$ deposition in 2014 for the different model runs relative to the default LOTOS-EUROS run. The first row indicates the changes related to the implementation of the updated $z_0$ values. The second row indicates the changes related to the implementation of the MODIS LAI values. The third row shows the combined effect of both these updates. (Please note the different scales).

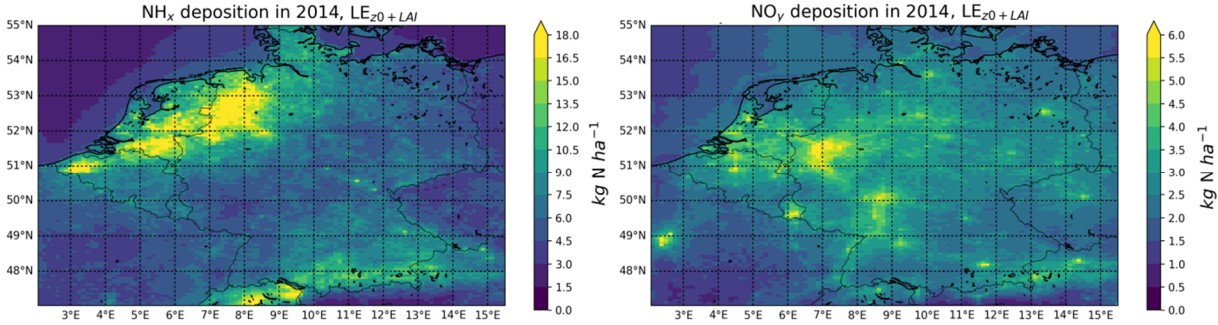


Figure 12: NH$_x$ deposition (left) and NO$_y$ deposition (right) in kg N ha$^{-1}$ in 2014.

Figure 13: The relative change in total NH$_x$ (top) and NO$_y$ (bottom) deposition in 2014 for the different model runs relative to the default LOTOS-EUROS run. The first row indicates the changes related to the implementation of the updated z$_0$ values. The second row indicates the changes related to the implementation of the MODIS LAI values. The third row shows the combined effect of both these updates.


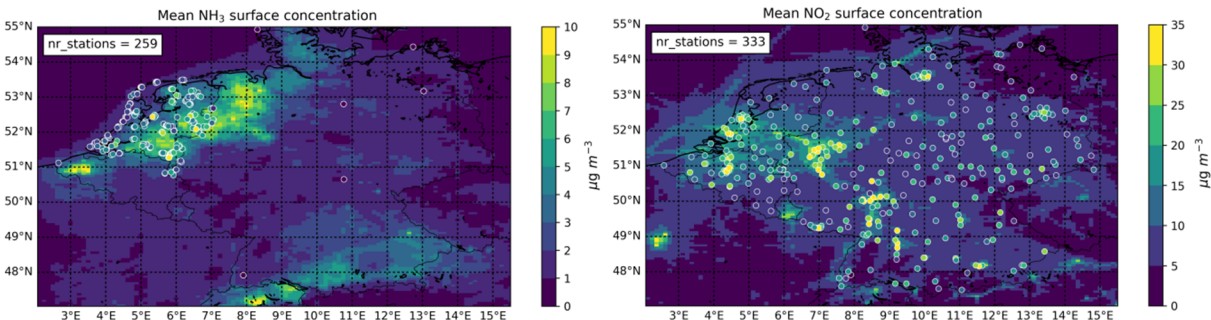

Figure 14: The yearly mean NH₃ (left) and NO₂ (right) surface concentrations in µg m⁻³ in 2014, and the corresponding mean surface concentrations measured at the in-situ stations.


Figure 15: The relative change (%) in mean NH₃ (left) and NO₂ (right) surface concentration in 2014 for the different model runs relative to the default LOTOS-EUROS run. The first row indicates the changes related to the implementation of the updated $z_0$ values. The second row indicates the changes related to the implementation of the MODIS LAI values. The third row shows the combined effect of both these updates.

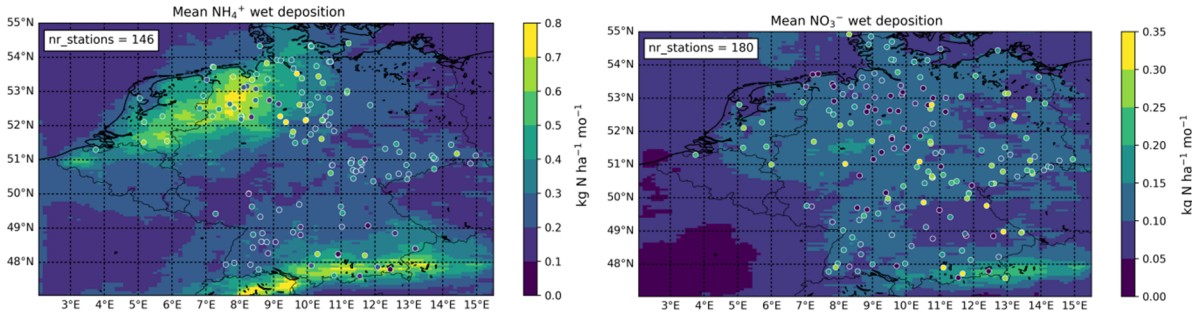

Figure 16: The mean NH$_4^+$ (left) and NO$_3^-$ (right) wet deposition in kg N ha$^{-1}$ mo$^{-1}$ in 2014.The mean observed wet deposition observed at the stations is plotted on top.

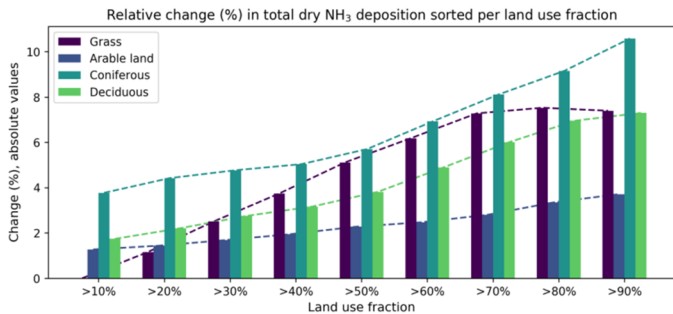

Figure 17: The relative difference (%) in total dry NH$_3$ deposition in 2014 between the default run (LE$_{default}$) and the run with the

 updated $z_0$ values (LE$_{z0}$), sorted by increasing land use fraction.

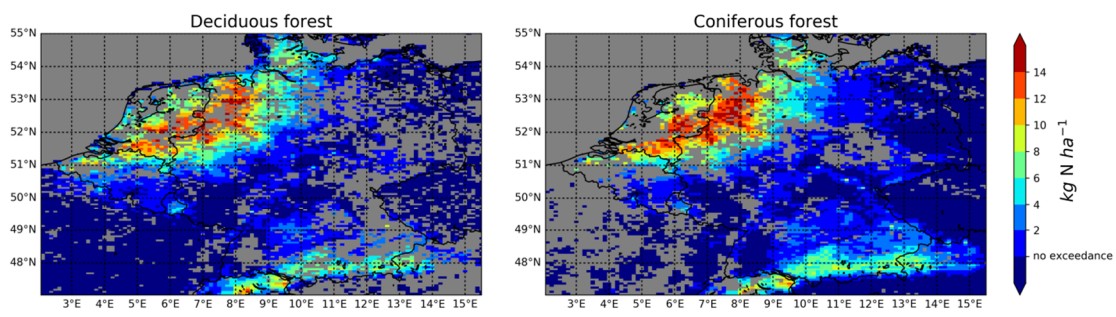

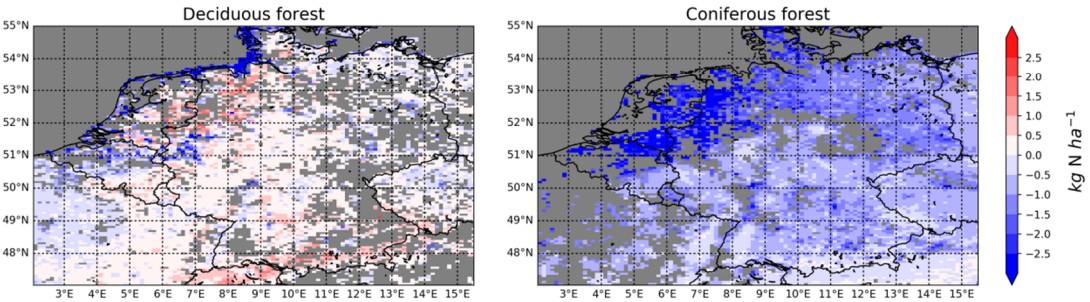

Figure 18: Critical load exceedances on forests in kg N ha$^{-1}$ in 2014. The upper figures show the critical on deciduous (left) and coniferous (right) forest, as modelled with the updated $z_0$ and LAI values. The lower figures show the absolute differences in

 critical load exceedance on deciduous (left) and coniferous (right) between the new and the default LOTOS-EUROS run.