# Peer review of "Satellite derived leaf area index and roughness length information for surface-atmosphere exchange modelling: a case study for reactive nitrogen deposition in north-western Europe using LOTOS-EUROS v2.0."

_Geoscientific Model Development, 2019_

## Short Comment (SC1) · 7 Nov 2019

Dear authors,

in my role as Executive editor of GMD, I would like to bring to your attention our Editorial version 1.2:

https://www.geosci-model-dev.net/12/2215/2019/

This highlights some requirements of papers published in GMD, which is also available

on the GMD website in the 'Manuscript Types' section:

http://www.geoscientific-model-development.net/submission/manuscript_types.html

In particular, please note that your paper is not of the type "Model experiment description paper" but it belongs in the category "Development and technical papers". As such it has to meet the following requirements:

- "If the model development relates to a single model then the model name and the version number must be included in the title of the paper. If the main intention of an article is to make a general (i.e. model independent) statement about the usefulness of a new development, but the usefulness is shown with the help of one specific model, the model name and version number must be stated in the title. The title could have a form such as, "Title outlining amazing generic advance: a case study with Model XXX (version Y)"."

- Code must be published on a persistent public archive with a unique identifier for the exact model version described in the paper or uploaded to the supplement, unless this is impossible for reasons beyond the control of authors. All papers must include a section, at the end of the paper, entitled "Code availability". Here, either instructions for obtaining the code, or the reasons why the code is not available should be clearly stated. It is preferred for the code to be uploaded as a supplement or to be made available at a data repository with an associated DOI (digital object identifier) for the exact model version described in the paper. Alternatively, for established models, there may be an existing means of accessing the code through a particular system. In this case, there must exist a means of permanently accessing the precise model version described in the paper. In some cases, authors may prefer to put models on their own website, or to act as a point of contact for obtaining the code. Given the impermanence of websites and email addresses, this is not encouraged, and authors should consider improving the availability with a more permanent arrangement. Making code available

through personal websites or via email contact to the authors is not sufficient. After the paper is accepted the model archive should be updated to include a link to the GMD paper.

Thus the title of you manuscript needs to include the model name and its version number. Additionally please ensure, that the exact model version you are using will be permanently available. Third, please note that the model code is not sufficient. As the most vital part of you analyses deals with the MODIS input data, the reader should be able to access these as well. Therefore please make the respective data sets available (or state which license issues needs to be fulfilled to get access).

Yours,

Astrid Kerkweg

---

## Referee Comment (RC1) · Anonymous Referee #1 · 17 Dec 2019

The authors present a modeling study in which z0 and LAI were replaced in a version of the LOTOS-EUROS model for Nitrogen deposition. They find that their changes to z0 and LAI have the potential to change local N deposition by up to 15% and find an integrated change in N deposition of ∼3-4%.

General/ major comments

1. In my opinion the paper reads more like a classical research article than a model development article. For a model development article, I would expect (i) a clear description of the changes (which is delivered), (ii) how these manifest in the underlying model equations (not clear as specific equations for R_a etc are missing), (iii) a thorough model evaluation/ validation against observations (which I don't think is extensive enough –> see further comments) and (iv) finally a brief application/ example of how these changes result in consequences in the environment (this is done extensively here). I am wondering whether this paper would be better suited in a different journal such as ACP, where the reviewers would focus more on the underlying science. At the same time more complete validation is needed.

2. Validation: I am a micrometerologist with experience in BVOC exchange and am thus not very familiar with the availability of N-data for validation purposes, however I feel that at present model validation is not sufficient. There is no attempt in the manuscript beyond Table 6 to evaluate whether values for z0 are reasonable. I note that values in this paper tend to be on the high side of other studies for forests and on the low side for grasslands. On that note, I also find it surprising that major urban centers are not seen at all in the z0 map (unlike coniferous forest (such as the black forest in SE Germany), which seem to have a very high z0's). z0 values could easily be determined from Fluxnet or other networks. z0 is linearly scaled with canopy height for forests which not very sophisticated. For croplands NDVI is also used for scaling, and I am wondering to what extent another vegetation index such as EVI, would be more appropriate as NDVI has problems in dense canopies. Similarly, LAI values should be sanity checked (see next point). Regarding observed N deposition and observed N observations: There is little evidence that the model presented here is an improvement. The full LAI and z0 model appears to have a worse RMSD and higher negative intercept, while R^2 values do not appear to change. I think that based on this evidence I would be hesitant to deploy the model on a larger scale. Regarding the improved N-species atmospheric concentrations: There are several potential reasons for mismatch between observed and modeled atmospheric N-species. First, the model underestimates N deposition (which may very well be the case) or second N-inventories underestimates emissions. For example U.S. methane emission inventories from oil and gas

are almost certainly 50% too low. Therefore, I think that the improved match between observed and modeled atmospheric N-species is not sufficient for model validation. In my opinion model validation should be as close as possible to the processes and state variables of the model as possible (e.g. deposition, resistance values, z0, and LAI). Also, I would expect a sensitivity study that shows for example deposition velocity V_d as a function of u, u*, LAI, z_0 or so. Without knowing the exact equations for the individual resistances (which should be in the paper) I find it hard to evaluate whether changing z0 and LAI independently of u* is sensible. Instead of section 4.5 just before the discussion I would expect a validation section at the beginning of the results.

3. Uncertainties: At no point in the paper is there any quantification of the uncertainties in modeled fluxes, measurements, or inventories. Given that the modified model in the mean produces a few percent of changes in N-deposition and LAI estimates are probably uncertain to at least 10% (similar arguments can be made for canopy heights) if not higher, I am wondering whether there is any significant difference in the newly modeled deposition.

4. LAI from MODIS and z0. There appears to be an unreasonably large seasonal cycle in MODIS derived coniferous LAI which for example then greatly affects results (e.g. Figure 7). Tian et al 2004 (10.1029/2003JD003777) have documented issues with Modis LAI during snow-covered conditions (and particularly affecting conifers) and care should be taken that this does not affect the very low LAI's presented here.

5. Application: I am not the best person to evaluate the application sections, but I feel that for a model development paper, this should be shortened (and potentially moved to an application paper), with this manuscript focusing mostly on model validation. I am providing limited comments on the text here, but feel that there may be too much conjecture based on the limited model validation that has been done here.

6. The manuscript is missing links to the code as needed for GMD.

Specific comments:

[Figure]

L53: N-emission inventories are mentioned here as uncertain, but are used for validation. I suggest to add more literature here that gives a quantitative estimate of the inventory uncertainty.

L60: "Most data assimilation and inversion methods rely on the assumption that sink terms in the model hold a negligible uncertainty." > This is probably true. However it is possible to create inversion systems that optimize more than one flux at the same time.

L87: "Under neutral conditions, the resulting logarithmic wind profile is defined as:" > Better: Under neutral conditions the resulting wind profile can be approximated using a logarithmic profile (or similar)

Sec 2.1.2: I would expect a more complete set of equations here for the resistances. Also the paper shows that z0 also affects wet deposition, but wet deposition is not described at all in the methods.

L192: "The pixels with percentages higher than 85% were isolated for each CORINE land cover class. We used the remaining pixels to compute z0 values for each CORINE land cover class. " > I don't understand what is done here.

Figure 1 and L218: "The equations are all within a reasonable range from one another for NDVI values lower than ∼0.8" > M90 and BS93 appear unreasonably high for NDVI >0.8. Please check whether these parameterizations remain valid for NDVI>0.8 or whether they should be removed before averaging.

L249: "The largest positive differences occur in forested areas, meaning that the default z0 values are lower than the updated z0 values. The largest negative deviations occur in urban areas and areas with "grass". " > z0 values for forests appear to be really high especially in southern Germany (>3 m). My guess would be that the simple scaling with canopy height fails here.

L264 and Figure 7: Should be "seasonal" not "yearly" variation. Also note the issue with coniferous LAI, which is clearly not estimated correctly in MODIS.

Figure 8: Not sure whether this figure is needed, these values could just be given in the text or a table. Also in my opinion model validation as mentioned above is needed here, before moving on to the case study.

Figure 4 & Figure 7: Why do these show only 4 out of 5 land use classes and the omitted class is different between figures?

Figure 13 & 15: It is hard to see how well these agree. In my opinion this is not enough for validation of the model, especially since Table 5 shows little to no effect on the aggregate deposition statistics and a worsening of concentration statistics. A more direct comparison by land use etc should be done here.

L344: "Of the two, the newly implemented MODIS LAI values seem to have a larger effect on both the NH3 and NO2 concentrations, leading to a slightly depreciated RMSE and slope for both NH3 and NO" > Please see my previous comment. I feel that issues with Modis LAI may be to blame here. Also I find it problematic that the only noticeable change in the error statistics is a worsening of performance, but statistics should be calculated for individual land use classes, given that grass/croplands and forests show dramatically different behavior.

Figure 16: Deposition is clearly sensitive to LAI and much less so to z0. However LAI from Modis is also very uncertain...

L370: "These differences can in part be explained by the occurrence of relatively tall forest canopy ($\sim$30 meters) in the dataset, especially in forest in southern Germany" > note that you are comparing your mean to the other studies and your estimated z0s for south German forests are clearly outside the range of the cited literature.

L384: "Here, however, we merely focus on updating the z0 values per land use, and we consider the effect of this beyond the scope of this paper" > I find this statement problematic because the changes of z0 may or may not be sensible on their own and there is no clear validation for them in this work.

L432: "This work has shown that changes in two of the main deposition parameters (LAI, z0) can already lead to distinct changes (∼30%) in the modelled deposition fields. " > The question remains though, whether these changes are justified

Technical: - Tables should have labels above the table. - brackets around citations are off when using (e.g.(). - Tables and figures should not be mixed L219 > Figure 2 > Fig 1? L239 > Into
* * *

---

## Referee Comment (RC2) · Anonymous Referee #2 · 18 Dec 2019

The work presented in this manuscript describes the results on nitrogen deposition fluxes from integrating spatially and temporally explicit values for z0 and LAI calculated from satellite-derived measurements. The paper reads really well, the description of the work done is very comprehensive and the figures informative.

It is regrettable that no prior sensitivity analyses of the model to LAI and z0 values was performed to determine whether implementing new parameterisations would expect to improve reactive N outputs (concentrations and fluxes). The authors reach this conclusion at the end of the study. I also find it a little ambiguous in grasping what is novel

in this paper. Clearly using satellite data for parametrising a CTM is but this does not improve drastically N deposition estimates. Perhaps a way around this is presenting this "new" methodology as a way to improve various exchange processes and argue that an example treated in this paper is the case of reactive N.

I have a few minor comments:

- Authors should better justify why they have chosen to take the average function of several different z0 functions for their models (Figure 1). Some pf those functions have similar trends and could be pooled together and averaged however others are different and a justification is needed.

- Authors should probably discuss or argument at the beginning the upside of using satellite derived data as compared to using outputs of biosphere models o from coupling LOTOS-EUROS with a biosphere model

- Better discussion is needed on the uncertainty related to reactive N deposition fluxes linked to (i) model parameterisation (Rcuticular, Rstomatal, gamma, . . .) but also sub-grid variability.

- A deeper discussion of the validity of assumptions (canopy height for forests, urban height, etc) and results of z0 and LAI compared to literature values, other models simulations but also to site scale measurements from networks (fluxnet, ICOS, . . .).

---

## Author Response (AR1)

Astrid Kerkweg

Dear authors, in my role as Executive editor of GMD, I would like to bring to your attention our Editorial version 1.2: https://www.geosci-model-dev.net/12/2215/2019/

This highlights some requirements of papers published in GMD, which is also available on the GMD website in the 'Manuscript Types' section: http://www.geoscientific-model-development.net/submission/manuscript_types.html

In particular, please note that your paper is not of the type "Model experiment description paper" but it belongs in the category "Development and technical papers". As such it has to meet the following requirements:

• "If the model development relates to a single model then the model name and the version number must be included in the title of the paper. If the main intention of an article is to make a general (i.e. model independent) statement about the usefulness of a new development, but the usefulness is shown with the help of one specific model, the model name and version number must be stated in the title. The title could have a form such as, "Title outlining amazing generic advance: a case study with Model XXX (version Y)"."

• Code must be published on a persistent public archive with a unique identifier for the exact model version described in the paper or uploaded to the supplement, unless this is impossible for reasons beyond the control of authors. All papers must include a section, at the end of the paper, entitled "Code availability". Here, either instructions for obtaining the code, or the reasons why the code is not available should be clearly stated. It is preferred for the code to be uploaded as a supplement or to be made available at a data repository with an associated DOI (digital object identifier) for the exact model version described in the paper. Alternatively, for established models, there may be an existing means of accessing the code through a particular system. In this case, there must exist a means of permanently accessing the precise model version described in the paper. In some cases, authors may prefer to put models on their own website, or to act as a point of contact for obtaining the code. Given the impermanence of websites and email addresses, this is not encouraged, and authors should consider improving the availability with a more permanent arrangement. Making code available through personal websites or via email contact to the authors is not sufficient. After the paper is accepted the model archive should be updated to include a link to the GMD paper.

Thus the title of you manuscript needs to include the model name and its version number. Additionally please ensure, that the exact model version you are using will be permanently available. Third, please note that the model code is not sufficient. As the most vital part of you analyses deals with the MODIS input data, the reader should be able to access these as well. Therefore please make the respective data sets available (or state which license issues needs to be fulfilled to get access).

Yours, Astrid Kerkweg

RC1: Interactive comment on "The effect of satellite derived leaf area index and roughness length information on modelled reactive nitrogen deposition in north-western Europe" by Shelley C. van der Graaf et al.

Anonymous Referee #1

The authors present a modeling study in which z0 and LAI were replaced in a version of the LOTOS-EUROS model for Nitrogen deposition. They find that their changes to z0 and LAI have the potential to change local N deposition by up to 15% and find an integrated change in N deposition of ∼3-4%.

General/ major comments
1. In my opinion the paper reads more like a classical research article than a model development article. For a model development article, I would expect (i) a clear description of the changes (which is delivered), (ii) how these manifest in the underlying model equations (not clear as specific equations for R_a etc are missing), (iii) a thorough model evaluation/ validation against observations (which I don't think is extensive enough –> see further comments) and (iv) finally a brief application/ example of how these changes result in consequences in the environment (this is done extensively here). I am wondering whether this paper would be better suited in a different journal such as ACP, where the reviewers would focus more on the underlying science. At the same time more complete validation is needed.

2. Validation: I am a micrometerologist with experience in BVOC exchange and am thus not very familiar with the availability of N-data for validation purposes, however I feel that at present model validation is not sufficient. There is no attempt in the manuscript beyond Table 6 to evaluate whether values for z0 are reasonable. I note that values in this paper tend to be on the high side of other studies for forests and on the low side for grasslands. On that note, I also find it surprising that major urban centers are not seen at all in the z0 map (unlike coniferous forest (such as the black forest in SE Germany), which seem to have a very high z0's). z0 values could easily be determined from Fluxnet or other networks. z0 is linearly scaled with canopy height for forests which not very sophisticated. For croplands NDVI is also used for scaling, and I am wondering to what extent another vegetation index such as EVI, would be more appropriate as NDVI has problems in dense canopies. Similarly, LAI values should be sanity checked (see next point). Regarding observed N deposition and observed N observations: There is little evidence that the model presented here is an improvement. The full LAI and z0 model appears to have a worse RMSD and higher negative intercept, while R^2 values do not appear to change. I think that based on this evidence I would be hesitant to deploy the model on a larger scale. Regarding the improved Nspecies atmospheric concentrations: There are several potential reasons for mismatch between observed and modeled atmospheric N-species. First, the model underestimates N deposition (which may very well be the case) or second N-inventories underestimates emissions. For example U.S. methane emission inventories from oil and gas are almost certainly 50% too low. Therefore, I think that the improved match between observed and modeled atmospheric N-species is not sufficient for model validation. In my opinion model validation should be as close as possible to the processes and state variables of the model as possible (e.g. deposition, resistance values, z0, and LAI). Also, I would expect a sensitivity study that shows for example deposition velocity V_d as a function of u, u*, LAI, z_0 or so. Without knowing the exact equations for the individual resistances (which should be in the paper) I find it hard to evaluate whether changing z0 and LAI independently of u* is sensible. Instead of section 4.5 just before the discussion I would expect a validation section at the beginning of the results.

3. Uncertainties: At no point in the paper is there any quantification of the uncertainties in modeled fluxes, measurements, or inventories. Given that the modified model in the mean produces a few percent of changes in N-deposition and LAI estimates are probably uncertain to at least 10% (similar arguments can be made for canopy heights) if not higher, I am wondering whether there is any significant difference in the newly modeled deposition.

4. LAI from MODIS and z0. There appears to be an unreasonably large seasonal cycle in MODIS derived coniferous LAI which for example then greatly affects results (e.g. Figure 7). Tian et al 2004 (10.1029/2003JD003777) have documented issues with Modis LAI during snow-covered conditions (and particularly affecting conifers) and care should be taken that this does not affect the very low LAI's presented here.

5. Application: I am not the best person to evaluate the application sections, but I feel that for a model development paper, this should be shortened (and potentially moved to an application paper), with this manuscript focusing mostly on model validation. I am providing limited comments on the text here, but feel that there may be too much conjecture based on the limited model validation that has been done here.

6. The manuscript is missing links to the code as needed for GMD.

Specific comments:
L53: N-emission inventories are mentioned here as uncertain, but are used for validation. I suggest to add more literature here that gives a quantitative estimate of the inventory uncertainty.

L60: "Most data assimilation and inversion methods rely on the assumption that sink terms in the model hold a negligible uncertainty." > This is probably true. However it is possible to create inversion systems that optimize more than one flux at the same time.

L87: "Under neutral conditions, the resulting logarithmic wind profile is defined as:" > Better: Under neutral conditions the resulting wind profile can be approximated using a logarithmic profile (or similar)

Sec 2.1.2: I would expect a more complete set of equations here for the resistances. Also the paper shows that z0 also affects wet deposition, but wet deposition is not described at all in the methods.

L192: "The pixels with percentages higher than 85% were isolated for each CORINE land cover class. We used the remaining pixels to compute z0 values for each CORINE land cover class. " > I don't understand what is done here.

Figure 1 and L218: "The equations are all within a reasonable range from one another for NDVI values lower than ~0.8" > M90 and BS93 appear unreasonably high for NDVI >0.8. Please check whether these parameterizations remain valid for NDVI>0.8 or whether they should be removed before averaging.

L249: "The largest positive differences occur in forested areas, meaning that the default z0 values are lower than the updated z0 values. The largest negative deviations occur in urban areas and areas with "grass". " > z0 values for forests appear to be really high especially in southern Germany (>3 m). My guess would be that the simple scaling with canopy height fails here.

L264 and Figure 7: Should be "seasonal" not "yearly" variation. Also note the issue with coniferous LAI, which is clearly not estimated correctly in MODIS.

Figure 8: Not sure whether this figure is needed, these values could just be given in the text or a table. Also in my opinion model validation as mentioned above is needed here, before moving on to the case study.

Figure 4 & Figure 7: Why do these show only 4 out of 5 land use classes and the omitted class is different between figures?

Figure 13 & 15: It is hard to see how well these agree. In my opinion this is not enough for validation of the model, especially since Table 5 shows little to no effect on the aggregate deposition statistics and a worsening of

concentration statistics. A more direct comparison by land use etc should be done here.

L344: "Of the two, the newly implemented MODIS LAI values seem to have a larger effect on both the NH3 and NO2 concentrations, leading to a slightly depreciated RMSE and slope for both NH3 and NO" > Please see my previous comment. I feel that issues with Modis LAI may be to blame here. Also I find it problematic that the only noticeable change in the error statistics is a worsening of performance, but statistics should be calculated for individual land use classes, given that grass/croplands and forests show dramatically different behavior.

Figure 16: Deposition is clearly sensitive to LAI and much less so to z0. However LAI from Modis is also very uncertain...

L370: "These differences can in part be explained by the occurrence of relatively tall forest canopy (~30 meters) in the dataset, especially in forest in southern Germany" > note that you are comparing your mean to the other studies and your estimated z0s for south German forests are clearly outside the range of the cited literature.

L384: "Here, however, we merely focus on updating the z0 values per land use, and we consider the effect of this beyond the scope of this paper" > I find this statement problematic because the changes of z0 may or may not be sensible on their own and there is no clear validation for them in this work.

L432: "This work has shown that changes in two of the main deposition parameters (LAI, z0) can already lead to distinct changes (~30%) in the modelled deposition fields. " > The question remains though, whether these changes are justified

Technical: - Tables should have labels above the table. - brackets around citations are off when using (e.g.(). - Tables and figures should not be mixed L219 > Figure 2 > Fig 1? L239 > Into

RC2: Interactive comment on "The effect of satellite derived leaf area index and roughness length information on modelled reactive nitrogen deposition in north-western Europe" by Shelley C. van der Graaf et al.

Anonymous Referee #2

The work presented in this manuscript describes the results on nitrogen deposition fluxes from integrating spatially and temporally explicit values for z0 and LAI calculated from satellite-derived measurements. The paper reads really well, the description of the work done is very comprehensive and the figures informative.

It is regrettable that no prior sensitivity analyses of the model to LAI and z0 values was performed to determine whether implementing new parameterisations would expect to improve reactive N outputs (concentrations and fluxes). The authors reach this conclusion at the end of the study. I also find it a little ambiguous in grasping what is novel in this paper. Clearly using satellite data for parametrising a CTM is but this does not improve drastically N deposition estimates. Perhaps a way around this is presenting this "new" methodology as a way to improve various exchange processes and argue that an example treated in this paper is the case of reactive N.

I have a few minor comments:
- Authors should better justify why they have chosen to take the average function of several different z0 functions for their models (Figure 1). Some pf those functions have similar trends and could be pooled together and averaged however others are different and a justification is needed.

 - Authors should probably discuss or argument at the beginning the upside of using satellite derived data as compared to using outputs of biosphere models o from coupling LOTOS-EUROS with a biosphere model - Better discussion is needed on the uncertainty related to reactive N deposition fluxes linked to (i) model parameterisation (Rcuticular, Rstomatal, gamma, . . .) but also subgrid variability.

- A deeper discussion of the validity of assumptions (canopy height for forests, urban height, etc) and results of z0 and LAI compared to literature values, other models simulations but also to site scale measurements from networks (fluxnet, ICOS, . . .).

**(2)    Author's response**

The authors would like to thank everyone for their valuable comments and contributions. We have responded to each of the comments individually in the following section. We first responded to each referee with a general response, followed (when necessary) by a more detailed reaction to some of the points stated by the referee.

**Response to SC1:**

We would like to thank Astrid Kerkweg for highlighting these issues, and we apologize for overlooking these requirements in the first place. We have changed the manuscript title and the data availability section following your comments. We changed the title of our manuscript to include the model name and version:
"Satellite derived leaf area index and roughness length information for surface-atmosphere exchange modelling: a case study for reactive nitrogen deposition in north-western Europe using LOTOS-EUROS v2.0."

We have changed the data availability section to:
"LOTOS-EUROS v2.0 is available for download under license at https://lotos-euros.tno.nl/. The adjusted model routines to read in the $z_0$ and LAI values are available in cooperation with TNO and can be disclosed upon request. All MODIS products used in this study are open-source and can be downloaded at  https://ladsweb.nascom.nasa.gov. The global forest canopy height dataset can be downloaded at https://daac.ornl.gov/. The population density grid can be downloaded at https://www.eea.europa.eu/."

**Response to RC1:**

We like to thank anonymous referee #1 for the time invested in this useful review and for his/her invaluable comments. We have changed multiple section in manuscript to make it clearer, mainly focusing on the underlying model equations such as a more complete description of the resistances. Also, we put extra emphasis on the underlying uncertainties in both the model and the measurements used in this study. We agreed that the validation part of the manuscript should be more thorough/complete and therefore added two main changes: 1) we split up the validation with in-situ measurement into different land use classes. 2) we directly looked at the derived $z_0$ values by comparing them to $z_0$ values computed from FLUXNET as suggested.

RC1: The authors present a modeling study in which z0 and LAI were replaced in a version of the LOTOS-EUROS model for Nitrogen deposition. They find that their changes to z0 and LAI have the potential to change local N deposition by up to 15% and find an integrated change in N deposition of ~3-4%.

General/ major comments
1. In my opinion the paper reads more like a classical research article than a model development article. For a model development article, I would expect (i) a clear description of the changes (which is delivered), (ii) how these manifest in the underlying model equations (not clear as specific equations for R_a etc are missing), (iii) a thorough model evaluation/ validation against observations (which I don't think is extensive enough –> see further comments) and (iv) finally a brief application/ example of how these changes result in consequences in the environment (this is done extensively here). I am wondering whether this paper would be better suited in a different journal such as ACP, where the reviewers would focus more on the underlying science. At the same time more complete validation is needed.
We decided to submit this manuscript to GMD, because we think that the method used in this paper could be useful for other model developers.

2. Validation: I am a micrometerologist with experience in BVOC exchange and am thus not very familiar with the availability of N-data for validation purposes, however I feel that at present model validation is not sufficient. There is no attempt in the manuscript beyond Table 6 to evaluate whether values for z0 are reasonable. I note that values in this paper tend to be on the high side of other studies for forests and on the low side for grasslands. On that note, I also find it surprising that major urban centers are not seen at all in the z0 map (unlike coniferous forest (such as the black forest in SE Germany), which seem to have a very high z0's). z0 values could easily be determined from Fluxnet or other networks.
We agree that it is hard to see the major urban centers in the composite $z_0$ map. Therefore, we have provided a figure with only the $z_0$ values for urban regions to the supplement. We have determined the $z_0$ values from several Fluxnet sites and compared the satellite-derived $z_0$ values to them. The results are described in section 4.3. It is indeed true that the $z_0$ values for grassland are a bit on the low side compared to the values used in other studies/models. However, we decided to use this function since the majority of the pixels classified as grassland in the Netherlands and Germany consist of managed grassland, which in general is shorter than (semi-) natural grasslands. The $z_0$ values of FLUXNET sites with short grass only also seem to match quite well to the satellite-derived values. But applied to other regions, this may result in overestimated $z_0$ values. We have added some lines to the discussion about this. In the future, a more detailed land cover classification that distinguishes natural from managed grasslands would be useful.

z0 is linearly scaled with canopy height for forests which not very sophisticated. For croplands NDVI is also used for scaling, and I am wondering to what extent another vegetation index such as EVI, would be more appropriate as NDVI has problems in dense canopies.
There are indeed multiple other studies available that link the $z_0$ values to other satellite derived vegetation product (e.g. EVI, LAI). We decided to use NDVI rather than EVI because we could find significantly more studies where

the link between NDVI and $z_0$ was derived. Moreover, the equations related to NDVI derived in these studies gave relatively similar outcomes for different crop types, which gave us more confidence for application over larger regions with various vegetation types. In a follow-up study, it would be a great idea to compare $z_0$ values derived with different vegetation product.

Similarly, LAI values should be sanity checked (see next point). Regarding observed N deposition and observed N observations: There is little evidence that the model presented here is an improvement. The full LAI and z0 model appears to have a worse RMSD and higher negative intercept, while R^2 values do not appear to change. I think that based on this evidence I would be hesitant to deploy the model on a larger scale. Regarding the improved Nspecies atmospheric concentrations: There are several potential reasons for mismatch between observed and modeled atmospheric N-species. First, the model underestimates N deposition (which may very well be the case) or second N-inventories underestimates emissions. For example U.S. methane emission inventories from oil and gas are almost certainly 50% too low.

The validation of the model results with in-situ measurements is indeed hampered by the uncertainties in N-inventories. Especially the emissions of $NH_3$, and as a consequence also the deposition, are indeed likely to be underestimated. We have added a paragraph in section 5 where we discuss the model uncertainty. It is true that this is but one aspect of the model uncertainty, and that there are several larger uncertainties to consider. This is indeed probably the main reason why validation with in-situ observations does not change drastically, However, we feel that it is also important to look at the contribution of the smaller model errors, especially if these lead to systematic changes in the resulting deposition distributions. In this paper we want to stress the fact that the sensitivity of our model to changes in LAI and $z_0$ values is large, and that changing these values can locally lead to significant differences compared to the model outcome before the changes. These changes can have consequences for the applications such as estimates of inputs related to ecosystem effects.

Therefore, I think that the improved match between observed and modeled atmospheric N-species is not sufficient for model validation. In my opinion model validation should be as close as possible to the processes and state variables of the model as possible (e.g. deposition, resistance values, z0, and LAI). Also, I would expect a sensitivity study that shows for example deposition velocity V_d as a function of u, u*, LAI, z_0 or so.

As mentioned above, we have added the $z_0$ values computed from FLUXNET to compare the satellite-derived $z_0$ values to.

Without knowing the exact equations for the individual resistances (which should be in the paper) I find it hard to evaluate whether changing z0 and LAI independently of u* is sensible. Instead of section 4.5 just before the discussion I would expect a validation section at the beginning of the results.

We have included an additional section in the LOTOS-EUROS model description where the other resistances are discussed, to give more insight into the effect of $z_0$ and LAI. The friction velocity u* in LOTOS-EUROS is directly computed from $z_0$ in the model, so it changes with changing $z_0$ values. The equation used to compute u* is now included in the model description.

3. Uncertainties: At no point in the paper is there any quantification of the uncertainties in modeled fluxes, measurements, or inventories. Given that the modified model in the mean produces a few percent of changes in N-deposition and LAI estimates are probably uncertain to at least 10% (similar arguments can be made for canopy heights) if not higher, I am wondering whether there is any significant difference in the newly modeled deposition.

We agree that a better description of the uncertainties is necessary. Therefore we extended the Dataset description section and added literature estimates of the model and measurement uncertainties.

4. LAI from MODIS and z0. There appears to be an unreasonably large seasonal cycle in MODIS derived coniferous LAI which for example then greatly affects results (e.g. Figure 7). Tian et al 2004 (10.1029/2003JD003777) have documented issues with Modis LAI during snow-covered conditions (and

particularly affecting conifers) and care should be taken that this does not affect the very low LAI's presented here. The MODIS-LAI values have been extensively validated in several different studies already (see Discussion), and therefore we chose not to put much more emphasis on the product itself in this paper. We did add two more sections about the MODIS-LAI: one where we compare the MODIS-LAI per FLUXNET site land use, and an additional section in the discussion about snow-covered conditions, as suggested. One other reasons for the large seasonal cycle in Figure 7 (now Figure 8) could be the mixing of land use classes. There are some discrepancies between the land use map used in the MODIS-LAI product and the land use map used in the model. Also, multiple corine land cover classes were grouped to create this figure. The result is that the average variation shown in this Figure is much more smoothened and does not clearly show inter-land use variations. Therefore, we added a new figure, Figure 9, where we show the MODIS-LAI values at FLUXNET sites with different land use classifications. In this Figure the differences between land use classes become clearer. Moreover, we checked whether our low winter-time LAI values for coniferous were related to snow-cover, but this was not the case. It is true that it could lead to considerable errors in frequently snow-covered regions, such as Scandinavia. These low LAI values should, however, have but a minor impact on the modelled deposition values. LOTOS-EUROS uses meteorological data to determine what regions are covered with snow. If a certain location is snow-covered, the parameterization for the canopy resistance is not used, instead an overall canopy resistance is used. Therefore, these low MODIS-LAI values do not affect the modelled deposition in LOTOS-EUROS during snow-cover.

5. Application: I am not the best person to evaluate the application sections, but I feel that for a model development paper, this should be shortened (and potentially moved to an application paper), with this manuscript focusing mostly on model validation. I am providing limited comments on the text here, but feel that there may be too much conjecture based on the limited model validation that has been done here.
We have shortened the application part of the paper by moving it partially to the supplement.

6. The manuscript is missing links to the code as needed for GMD.
We have added an additional statement to the "data availability" statement.

Specific comments:
L53: N-emission inventories are mentioned here as uncertain, but are used for validation. I suggest to add more literature here that gives a quantitative estimate of the inventory uncertainty.
We have added an additional section to the discussion about the model uncertainties.

L60: "Most data assimilation and inversion methods rely on the assumption that sink terms in the model hold a negligible uncertainty." > This is probably true. However it is possible to create inversion systems that optimize more than one flux at the same time.
To optimize more than one flux, they should have a different impact on the observed values in space and/or time. In chemical transport models, however, usually (satellite-observed) concentrations are assimilated. As the emissions and the deposition terms both directly impact the concentrations, it is difficult to optimize both.

L87: "Under neutral conditions, the resulting logarithmic wind profile is defined as:" > Better: Under neutral conditions the resulting wind profile can be approximated using a logarithmic profile (or similar)
Thank you, we changed the line as suggested.

Sec 2.1.2: I would expect a more complete set of equations here for the resistances. Also the paper shows that z0 also affects wet deposition, but wet deposition is not described at all in the methods.
We have added a section describing the other equations for the resistances. Since the effect in the wet deposition is only indirect, due to changes in the dry deposition and consequently the atmospheric concentrations, we have decided to leave this out.

L192: "The pixels with percentages higher than 85% were isolated for each CORINE land cover class. We used the remaining pixels to compute z0 values for each CORINE land cover class. " > I don't understand what is done here. We have rewritten this part of the description.

Figure 1 and L218: "The equations are all within a reasonable range from one another for NDVI values lower than ~0.8" > M90 and BS93 appear unreasonably high for NDVI >0.8. Please check whether these parameterizations remain valid for NDVI>0.8 or whether they should be removed before averaging.
Equation M90 and BS93 are indeed high for NDVI > 0.8. However, since only a small percentage of the NDVI values in our study area are above 0.8, it does not pose a problem for our study area. We have some additional explanation to this section as well as an extra figure to the supplement to illustrate this.

L249: "The largest positive differences occur in forested areas, meaning that the default z0 values are lower than the updated z0 values. The largest negative deviations occur in urban areas and areas with "grass". " > z0 values for forests appear to be really high especially in southern Germany (>3 m). My guess would be that the simple scaling with canopy height fails here.
It is indeed true that the $z_0/h$ ratio for very tall trees may be smaller. This is for instance the case for some tropical forests. Our current study area, however, consists of mostly temperate forests. Nakai et al., 2008 determines the $z_0/h$ value for different types of boreal and temperate forest with varying forest canopy heights and found that it is more or less constant and ~0.1. We also looked at the distribution of the forest canopy heights (see Figure S1), to check the amount of relatively tall trees are in our study area. We found that the majority of the forest canopy heights in the modelled region are below 30 meters. In other regions it would be a good idea to set a maximum $z_0$ value to any avoid issues.

L264 and Figure 7: Should be "seasonal" not "yearly" variation. Also note the issue with coniferous LAI, which is clearly not estimated correctly in MODIS.
We have changed this and we have extended the section in the discussion part of the paper regarding the uncertainty of the MODIS-LAI values (see response to point 4).

Figure 8: Not sure whether this figure is needed, these values could just be given in the text or a table. Also in my opinion model validation as mentioned above is needed here, before moving on to the case study.
We moved this Figure to the supplement.

Figure 4 & Figure 7: Why do these show only 4 out of 5 land use classes and the omitted class is different between figures?
We did not show the roughness lengths for coniferous here, since they have a fixed value throughout the year. We omitted permanent crops in the other figure because the LAI values in the model are the same as those for arable land. We decided to remove Figure 4 (the roughness lengths) altogether and replace it with Figure 5. In this Figure the seasonal variation of the roughness lengths for croplands are shown.

Figure 13 & 15: It is hard to see how well these agree. In my opinion this is not enough for validation of the model, especially since Table 5 shows little to no effect on the aggregate deposition statistics and a worsening of concentration statistics. A more direct comparison by land use etc should be done here.
We have matched the land use map with the location of the in-situ measurements and computed these statistics per land use class. The results are plotted in Table S2.

L344: "Of the two, the newly implemented MODIS LAI values seem to have a larger effect on both the NH3 and NO2 concentrations, leading to a slightly depreciated RMSE and slope for both NH3 and NO" > Please see my previous comment. I feel that issues with Modis LAI may be to blame here. Also I find it problematic that the only noticeable change in the error statistics is a worsening of performance, but statistics should be calculated for

individual land use classes, given that grass/croplands and forests show dramatically different behavior.

Please see the previous comment. Table S2 shows the r and RMSD split up per land use class. There are some inter-land use differences, but in general we also observe a slight depreciation in the RMSD when including the MODIS-LAI, especially for the $NH_3$ and $NO_2$ surface concentrations. In the new version of the manuscript, we try to put more emphasis on the fact that these MODIS-LAI are not perfect, and do not necessarily lead to much better model performance, they are but another possible option as input data for the LAI. There are other CTMs (for instance GEOS-Chem) that currently use these values. And we should thus be mindful that the choice of input LAI values already leads to clear redistributions and structural changes in the modelled deposition fields.

Figure 16: Deposition is clearly sensitive to LAI and much less so to z0. However LAI from Modis is also very uncertain...

L370: "These differences can in part be explained by the occurrence of relatively tall forest canopy (~30 meters) in the dataset, especially in forest in southern Germany" > note that you are comparing your mean to the other studies and your estimated z0s for south German forests are clearly outside the range of the cited literature.

The $z_0$ values are indeed on the high side compared to literature studies. On the other hand, the newly added $z_0$ estimates from FLUXNET show a similar range.

L384: "Here, however, we merely focus on updating the z0 values per land use, and we consider the effect of this beyond the scope of this paper" > I find this statement problematic because the changes of z0 may or may not be sensible on their own and there is no clear validation for them in this work.

We changed this statement in the paper.

L432: "This work has shown that changes in two of the main deposition parameters (LAI, z0) can already lead to distinct changes (~30%) in the modelled deposition fields. " > The question remains though, whether these changes are justified.

Technical: - Tables should have labels above the table. - brackets around citations are off when using (e.g.(). - Tables and figures should not be mixed L219 > Figure 2 > Fig 1? L239 > Into

Thank you, we changed this.

**Response to RC2:**

The authors greatly thank anonymous referee #2 for taking the time to review our manuscript and his/her useful comments, and we changed the manuscript accordingly. The main changes in the manuscript include a more thorough discussion of the model uncertainties, and an extra section where we validate our $z_0$ estimates with $z_0$ values computed from FLUXNET sites.

RC2: The work presented in this manuscript describes the results on nitrogen deposition fluxes from integrating spatially and temporally explicit values for z0 and LAI calculated from satellite-derived measurements. The paper reads really well, the description of the work done is very comprehensive and the figures informative.

It is regrettable that no prior sensitivity analyses of the model to LAI and z0 values was performed to determine whether implementing new parameterisations would expect to improve reactive N outputs (concentrations and fluxes). The authors reach this conclusion at the end of the study. I also find it a little ambiguous in grasping what is novel in this paper. Clearly using satellite data for parametrising a CTM is but this does not improve drastically N deposition estimates. Perhaps a way around this is presenting this "new" methodology as a way to improve various exchange processes and argue that an example treated in this paper is the case of reactive N.
We agree, and therefore we changed the title of the manuscript to: "Satellite derived leaf area index and roughness length information for surface-atmosphere exchange modelling: a case study for reactive nitrogen deposition in north-western Europe using LOTOS-EUROS v2.0."

I have a few minor comments:
- Authors should better justify why they have chosen to take the average function of several different z0 functions for their models (Figure 1). Some pf those functions have similar trends and could be pooled together and averaged however others are different and a justification is needed.
We have added some extra explanation to this section, as well as an extra figure. It is indeed true that some of the equations deviate from one another. This seems to be especially the case for larger NDVI values (>0.8) . We have computed the NDVI value in our study area, and almost all values are below 0.8 year-round. Within this range, the average function does not differ that much from the individual functions. Another reason why we chose to use the average function for the different crop types (categories: arable land, other, and permanent crops) is because we do not have an accurate map of the different crop types for our study area. Even with a good crop map it would be difficult to decide on what function to use, since the relation between $z_0$ and NDVI is not studied for all crop types.

 - Authors should probably discuss or argument at the beginning the upside of using satellite derived data as compared to using outputs of biosphere models o from coupling LOTOS-EUROS with a biosphere model - Better discussion is needed on the uncertainty related to reactive N deposition fluxes linked to (i) model parameterisation (Rcuticular, Rstomatal, gamma, . . .) but also subgrid variability.
We added some lines about LAI from biosphere models to the introduction of the paper. We added a paragraph about the model uncertainties to the discussion, also including uncertainties related to deposition parameterization.

- A deeper discussion of the validity of assumptions (canopy height for forests, urban height, etc) and results of z0 and LAI compared to literature values, other models simulations but also to site scale measurements from networks (fluxnet, ICOS, . . .).
We have extended the discussion about the LAI values, and also added an extra figure (Figure 9). We moved the table with literature values of $z_0$ to section 4.2. and added a section about the comparison to $z_0$ values at FLUXNET sites. For some of the FLUXNET forest sites we found an estimate of the maximum forest canopy height (Flechard et al., 2019). For all but one site (DE-HAI) the forest canopy height from GLAS are lower than this value. We added this comparison to the supplementary material and discuss it shortly in the discussion of the manuscript.

**(3)     Author's changes to manuscript**

The lines that are added or changed are marked in green. The lines that have been removed are marked in red.
* * *

[revised manuscript text omitted]

Relative change (%) in mean $NH_3$ surface concentration in 2014      Relative change (%) in mean $NO_2$ surface concentration in 2014

Figure 15: The relative change (%) in mean $NH_3$ (left) and $NO_2$ (right) surface concentration in 2014 for the different model runs relative to the default LOTOS-EUROS run. The first row indicates the changes related to the implementation of the updated $z_0$ values. The second row indicates the changes related to the implementation of the MODIS LAI values. The third row shows the combined effect of both these updates.

[Figure]

Figure 16: The mean $NH_4^+$ (left) and $NO_3^-$ (right) wet deposition in kg N ha$^{-1}$ mo$^{-1}$ in 2014.The mean observed wet deposition observed at the stations is plotted on top.

[Figure]

Figure 17: The relative difference (%) in total dry $NH_3$ deposition in 2014 between the default run (LE$_{default}$) and the run with the updated $z_0$ values (LE$_{z0}$), sorted by increasing land use fraction.

[Figure]

Figure 18: Critical load exceedances on forests in kg N ha$^{-1}$ in 2014. The upper figures show the critical on deciduous (left) and coniferous (right) forest, as modelled with the updated $z_0$ and LAI values. The lower figures show the absolute differences in critical load exceedance on deciduous (left) and coniferous (right) between the new and the default LOTOS-EUROS run.

---

## Author Response (AR2)

Topical Editor

Dear Authors,

I have now received the reviews on your updated manuscript. Please address the suggestions and recommendations made by referee#1. Moreover, please ensure you comply with the requirements necessary to GMD as highlighted by the reviewer.

your sincerely,

Jason Williams

Dear Jason Williams,

Thank you for your response. We have made some additional changes in the manuscript following the remarks by referee #1. The changes made to the manuscript are included below, parts that are removed are red and parts that are added are shown in green. Regarding the GMD requirements, we once more looked at the LOTOS-EUROS v2.0 open source license from TNO. The LOTOS-EUROS v2.0 code used in this study is available for anyone to download, provided you fill out an application form once. However, the users license states that it is prohibited to re-distribute the model code to third parties. The complete model base code, is however, freely accessible and stored on the website given in the code availability section. We made a minor change in the data availability section and with this we hope to fulfill the GMD requirements. If there is any need to add or change anything, please let us know.

Kind regards,
On behalf of all authors,

Shelley van der Graaf

Thank you for your comments, they were very helpful. We made some changes to the manuscript accordingly, which we will discuss below. Regarding the GMD requirements, we carefully looked at the LOTOS-EUROS v2.0 open source user license, and unfortunately we are not allowed to share the model base code with third parties outside TNO. However, the full model code is freely available online and stored the website mentioned in the "code availability" section, provided you fill in an application form once. The adjustment made to LOTOS-EUROS are very simple and only include small changes in the input data routines. Most programmers can easily make these adjustments, but just in case we will also store these adjusted routines on TNOs server and make them available upon request. We changed the code availability statement to:

"LOTOS-EUROS v2.0 is open source and available for download under license at https://lotos-euros.tno.nl/. The adjusted model routines to read in the $z_0$ and LAI values are available in cooperation with TNO and can be disclosed upon request."

I would like to thank the authors for their work and their response to my comments. I do think that the efforts in this paper are worthy of publication (with some revisions). I am still not convinced that this paper fulfills all requirements for a GMD paper (such as all code needs to be available via 3rd party stable repository). I still believe the paper may be better suited for ACP rather than GMD.

I have the following additional comments:

Sec 2.1.2: The parameterization for L should be cited. I understand that z/L is very hard to obtain and that this is part of the original model, but I am wondering what the sensitivity of errors in z/L is compared to errors in z0 or LAI? (This is just a comment and does not need to be addressed)

We have added a citation for the parameterization for L.

Sec 3.3. Thanks for doing this analysis. Based on this your original results seem sensible. i am wondering whether this could be put in the supplement, rather than here?

As suggested, we moved this part of the methodology to the supplement.

One thing to mention would be though that z0 from Fluxnet is both due to the vegetation influence and topography, while your scaling is only due to canopy height. In the end, since forests are often in complex terrain, this does not seem to matter much (at least in this case), but I would mention this in the text.

Sec 4.2: See my comment to Sec 3.3. I think that there is covariance between mountains (i.e. black forest) and tree heights. These mountains contribute to z0 in addition to the tree height effect alone. I am not sure how to do this better and results seem reasonable, so I would just discuss this briefly here.

We agree that both these statements are probably true and we have added some lines about it to section 4.3 as well as to the discussion part of the article.

Figure 5: It is not quite clear what is modeled and what is observed from the legend

Thank you, we added some lines to the Figure description to clarify this.

Figure 9: It would be better to plot individual sites here rather than mean and std. One could add the mean as well. In general F

We changed the Figure as suggested, and moved the original one to the supplement.

Some of the language in the newly written sections could require some editing.

To improve the new parts of the manuscript, we asked a colleague to proofread it. A number of sentences were changes, we marked those in red in the manuscript changes document.

[revised manuscript text omitted]